# TGPO: Temporal Grounded Policy Optimization for Signal Temporal Logic Tasks

## Abstract

Learning control policies for complex, long-horizon tasks is a central challenge in robotics and autonomous systems. Signal Temporal Logic (STL) offers a powerful and expressive language for specifying such tasks, but its non-Markovian nature and inherent sparse reward make it difficult to be solved via standard Reinforcement Learning (RL) algorithms. Prior RL approaches focus only on limited STL fragments or use STL robustness scores as sparse terminal rewards. In this paper, we propose TGPO, Temporal Grounded Policy Optimization, to solve complex STL tasks. TGPO decomposes STL into timed subgoals and invariant constraints and provides a hierarchical framework to tackle the problem. The high-level component of TGPO proposes concrete time allocations for these subgoals, and the low-level time-conditioned policy learns to achieve the sequenced subgoals using a dense, stage-wise reward signal. During inference, we sample various time allocations and select the most promising assignment for the policy network to rollout the solution trajectory. To foster efficient policy learning for complex STL with multiple subgoals, we leverage the learned critic to guide the high-level temporal search via Metropolis-Hastings sampling, focusing exploration on temporally feasible solutions. We conduct experiments on five environments, ranging from low-dimensional navigation to manipulation, drone, and quadrupedal locomotion. Under a wide range of STL tasks, TGPO significantly outperforms state-of-the-art baselines (especially for high-dimensional and long-horizon cases), with an average of 31.6% improvement in task success rate compared to the best baseline.

## 1 Introduction

Signal Temporal Logic (STL) is a powerful framework for specifying tasks with temporal and spatial constraints in real-world robotic applications. However, designing controllers to satisfy these specifications is difficult, especially for systems with complex dynamics and a long task horizon. While Reinforcement Learning (RL) excels in handling these dynamical systems, directly deploying RL for STL specifications poses significant challenges. The history-dependent nature of STL breaks the Markovian assumption for the common RL algorithms. Furthermore, the reward based on the STL satisfaction is extremely sparse for long-horizon tasks, making RL struggle to learn effectively.

Existing model-free RL approaches for STL tasks typically leverage state augmentation with reward shaping. $\tau$-MDP (Aksaray et al., 2016) encodes histories explicitly in the augmented spaces and F-MDP (Venkataraman et al., 2020) designs flags to bookkeep the satisfaction of STL subformulas. However, these techniques only work on limited STL fragments with up to two temporal layers. While model-based RL (Kapoor et al., 2020; He et al., 2024) has fewer restrictions on the STL formulas, learning the system (latent space) dynamics can be challenging, and the estimation error accumulates over long horizons. Additionally, the planning often relies on Monte Carlo Tree Search or sampling action sequences, which may not be tractable for high-dimensional systems.

We argue that the primary barrier for RL to efficiently solve STL tasks is the difficulty of designing a dense, stage-wise reward function. This challenge stems directly from the unspecified temporal variables governing the "reach"-type tasks in STL formulas, which prevents a direct decomposition of STL into a sequence of executable subgoals. For example, for an STL $F_{[0,160]}A \wedge F_{[0,160]}B$ ("Eventually reach $A$ and eventually reach $B$ within the time interval $[0, 160]$"), the time assignments for reaching $A$ and reaching $B$ determine the order of visiting these regions. If we can ground the

variables into concrete values (e.g., reach $A$ at 35, and reach $B$ at 120), the problem can be cast into a sequence of goal-reaching problems, which is much easier to solve by RL.

Inspired by this observation, we propose a hierarchical RL framework to solve STL tasks by iteratively conducting **T**emporal **G**rounding and **P**olicy **O**ptimization (TGPO). The high-level component assigns values for the time variables to form the sequenced subgoals, and the low-level time-conditioned policy learns to achieve the task guided by the dense, stage-wise rewards derived from these subgoals. To efficiently bind values for multiple time variables, we carry out a high-level temporal search with a critic that predicts STL satisfaction. A Metropolis–Hastings sampling is used to guide exploration toward more "promising" time allocations. During inference, we sample time variable assignments and evaluate them using the critic. The most promising schedule is then executed by the low-level policy to generate the final solution trajectory for the STL specification.

We conduct extensive experiments over five simulation environments, ranging from 2D linear dynamics to 29D Ant navigation tasks. Compared to other baselines, TGPO[*] (with Bayesian time variable sampling) achieves the highest overall task success rate. The performance gains are significant, especially in high-dimensional systems and long-horizon tasks. Furthermore, our time-conditioned design offers key benefits: our critic offers interpretability by identifying promising temporal plans, and the policy can generate diverse, multi-modal behaviors to satisfy a single STL specification.

Our main contributions are summarized as follows: (1) **Hierarchical RL-STL framework**: To the best of our knowledge, we are the first to develop a hierarchical model-free RL algorithm capable of solving deeply-nested, finite-time STL tasks over long horizons. (2) **Critic-guided Bayesian sampling**: We introduce a critic-guided temporal grounding mechanism that, together with STL decomposition, yields subgoals and invariant constraints. This mechanism constructs an augmented MDP with dense, stage-wise rewards and thus overcomes the sparse reward challenges that have hindered existing RL approaches. (3) **Interpretability**: By explicitly grounding subgoals and invariant constraints in the STL structure using critic-guided Bayesian sampling, our approach offers a more interpretable learning process, where progress can be directly traced to logical task components. (4) **Complex dynamics and reproducibility**: TGPO demonstrates strong performance over other baselines and fits for complex dynamics, which supports the effectiveness of the design. All the code (the algorithm, the simulations and STL tasks) will be open-sourced to advance STL planning.

## 2 RELATED WORK

### 2.1 SIGNAL TEMPORAL LOGIC TASKS

Signal Temporal Logic (STL) offers a powerful framework for specifying robotics tasks (Donzé, 2013). Unlike Linear Temporal Logic (LTL), STL operates over continuous signals with time intervals and lacks an automaton representation, making it challenging to conduct planning (Finucane et al., 2010). Traditional approaches for STL include sampling-based methods (Vasile et al., 2017; Karlsson et al., 2020; Linard et al., 2023; Sewlia et al., 2023), Mixed-integer Programming (Sun et al., 2022; Kurtz & Lin, 2022) and trajectory optimization (Leung et al., 2023). More recently, learning-based methods emerged, such as differentiable policy learning (Liu et al., 2021; 2023; Meng & Fan, 2023), imitation learning (Puranic et al., 2021; Leung & Pavone, 2022; Meng & Fan, 2024; 2025), and reinforcement learning (RL) (Liao, 2020).

### 2.2 REINFORCEMENT LEARNING FOR TEMPORAL LOGIC TASKS

Temporal logic RL has been extensively studied in Linear Temporal Logic (LTL) and some Signal Temporal Logic (STL) fragments (Liao, 2020), where the key challenge is designing suitable rewards. For LTL, existing methods (Sadigh et al., 2014; Li et al., 2017; Hasanbeig et al., 2018; 2020) typically convert the formula into Limit-Deterministic Büchi Automata (LDBA) (Sickert et al., 2016) or reward machines (Icarte et al., 2018), while LTL2Action (Vaezipoor et al., 2021) uses progression (Bacchus & Kabanza, 2000) to assign dense reward, and SpectRL (Jothimurugan et al., 2019) devises a composable specification language for complex objectives. In contrast, STL poses additional challenges due to its explicit time constraints and real-value predicates. Early approaches augment the state space via temporal abstractions using history segments (Aksaray et al., 2016; Ikemoto & Ushio, 2022) or flags (Venkataraman et al., 2020; Wang et al., 2024), while bounded horizon nominal robustness (BHNR) (Balakrishnan & Deshmukh, 2019) offers intermediate reward

approximations. Recent work uses model-based learning to solve STL tasks with evolutionary strategies (Kapoor et al., 2020) and Monte-Carlo Tree Search in value function space (He et al., 2024). However, most of these methods are restricted to STL structures and systems (limited temporal nesting, fixed-size time windows, or grid-like environments). Instead, our method can handle more complex STLs and efficiently designs augmented states along with dense, stage-wise rewards.

## 3 PRELIMINARIES

### 3.1 SIGNAL TEMPORAL LOGIC (STL)

Consider a discrete-time system $x_{t+1} = f(x_t, u_t)$ where $x_t \in \mathcal{X} \subseteq \mathbb{R}^n$ and $u_t \in \mathcal{U} \subseteq \mathbb{R}^m$ denote the state and control at time $t$. Starting from an initial state $x_0$, a signal $\sigma = x_0, ..., x_T$ is generated via controls $u_0, ..., u_{T-1}$. STL specifies properties via the following rules (Donzé et al., 2013):

$$\phi ::= \top \mid \mu(x) \geq 0 \mid \neg\phi \mid \phi_1 \wedge \phi_2 \mid \phi_1 U_{[a,b]}\phi_2. \tag{1}$$

Here the boolean-type operators split by "|" are the building blocks to compose an STL: $\top$ means "true", $\mu$ denotes a predicate function $\mathbb{R}^n \to \mathbb{R}$, and $\neg, \wedge, U, {}_{[a,b]}$ are "negation", "conjunction", "until" and the time interval from $a$ to $b$. Other operators are "disjunction": $\phi_1 \vee \phi_2 = \neg(\neg\phi_1 \wedge \neg\phi_2)$, "eventually": $F_{[a,b]}\phi = \top U_{[a,b]}\phi$ and "always": $G_{[a,b]}\phi = \neg F_{[a,b]}\neg\phi$. We denote $\sigma, t \models \phi$ if the signal $\sigma$ from time $t$ satisfies the STL formula (the evaluation of $\phi$ returns True). In particular, we simply write $\sigma \models \phi$ if the signal is evaluated from $t = 0$. For operators $\top, \mu \geq 0, \neg, \wedge$ and $\vee$, the evaluation checks for the signal state at time $t$. As for temporal operators (Maler & Nickovic, 2004): $\sigma, t \models F_{[a,b]}\phi \Leftrightarrow \exists t' \in [t+a, t+b], \sigma, t' \models \phi$; and $\sigma, t \models G_{[a,b]}\phi \Leftrightarrow \forall t' \in [t+a, t+b], \sigma, t' \models \phi$; and $\sigma, t \models \phi_1 U_{[a,b]}\phi_2 \Leftrightarrow \exists t' \in [t + a, t + b], \sigma, t' \models \phi_2, \forall t'' \in [0, t'], \sigma, t'' \models \phi_1$. In plain words, $\phi_1 U_{[a,b]}\phi_2$ means "$\phi_1$ holds until $\phi_2$ happens in $[a,b]$." Robustness score (Donzé & Maler, 2010) $\rho(\sigma, t, \phi)$ measures how well a signal $\sigma$ satisfies $\phi$. We have $\rho \geq 0$ iff $\sigma, t \models \phi$. The score $\rho$ is:

$$\begin{cases} \rho(\sigma, t, \top) = 1 \\ \rho(\sigma, t, \mu) = \mu(\sigma(t)) \\ \rho(\sigma, t, \neg\phi) = -\rho(\sigma, t, \phi) \\ \rho(\sigma, t, \phi_1 \wedge \phi_2) = \min\left(\rho(\sigma, t, \phi_1), \rho(\sigma, t, \phi_2)\right) \\ \rho(\sigma, t, F_{[a,b]}\phi) = \sup_{r \in [a,b]} \rho(\sigma, t + r, \phi) \\ \rho(\sigma, t, G_{[a,b]}\phi) = \inf_{r \in [a,b]} \rho(\sigma, t + r, \phi) \\ \rho(\sigma, t, \phi_1 U_{[a,b]}\phi_2) = \sup_{t' \in [t+a, t+b]} \min\left\{\rho(\sigma, t', \phi_2), \inf_{t'' \in [t, t']} \rho(\sigma, t'', \phi_1)\right\} \end{cases} \tag{2}$$

In this work, we do not consider STLs with disjunctions or temporal structures of the form $GF(...)$.

### 3.2 MARKOV DECISION PROCESS

A Markov Decision Process (MDP) is defined by the tuple $\mathcal{M} = (\mathcal{S}, \mathcal{A}, P, R, \gamma)$ where: $\mathcal{S}$ and $\mathcal{A}$ represent the sets of states and actions, respectively, $P : \mathcal{S} \times \mathcal{A} \times \mathcal{S} \to [0, 1]$ is the probabilistic transition function where $P(s'|s, a)$ denotes the probability of the next state $s'$ given current state $s$ and action $a$, $R : \mathcal{S} \times \mathcal{A} \to \mathbb{R}$ is the reward function, and $\gamma \in [0, 1)$ is the discount factor. The agent decision is made by a policy $\pi : \mathcal{S} \to \mathcal{A}$ which maps states to a probability distribution over actions. The objective is to find an optimal policy $\pi^*$ that maximizes the expected discounted cumulative reward from a starting state $s_0$: $\mathbb{E}_\pi\left[\sum_{t=0}^{\infty} \gamma^t R(s_t, a_t)\big|s_0\right]$ with $a_t \sim \pi(\cdot|s_t)$ and $s_{t+1} \sim P(\cdot|s_t, a_t)$.

### 3.3 PROBLEM FORMULATION

Consider a discrete-time system with state space $\mathcal{X}$, control space $\mathcal{U}$ and the initial state distribution $\rho_0$ over the support $\mathcal{X}_0 \subseteq \mathcal{X}$. Given an STL $\phi$ defined in Eq. 1, our objective is to learn a policy $\pi$ that can generate a trajectory $\tau$ to maximize the STL satisfaction probability: $\max_\pi \mathbb{P}_{x_0 \sim \rho_0} (\tau \models \phi)$.

**Remarks.** It is tempting to directly solve it via MDP, treating the control system state $\mathcal{X}$ as the MDP state $\mathcal{S}$, and the control input $\mathcal{U}$ as the action space $\mathcal{A}$. However, for STL tasks, the policy also depends on the history[1], making the problem non-Markovian. Thus, we need to augment the MDP state to keep history data. Besides, the satisfaction of an STL is checked over the full trajectory, making it difficult to define dense rewards (unlike LTL, where stage-wise rewards (Camacho et al., 2017; Vaezipoor et al., 2021) can be defined). Thus, we need to design dense rewards under the augmented state space to learn efficiently.

## 4 METHODOLOGY

We propose TGPO, Temporal Grounded Policy Optimization, to address the problem considered. The entire framework is illustrated in Fig. 1, and we explain each component in detail below.

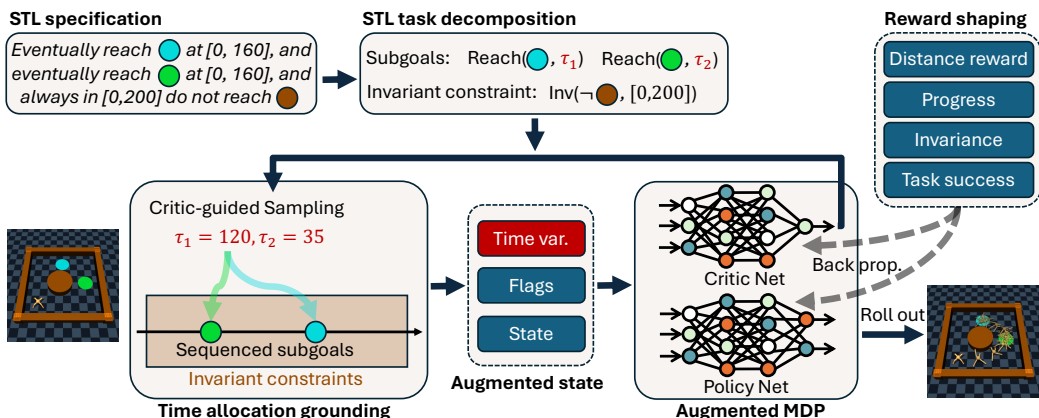

Figure 1: Framework: STL decomposition and critic-guided temporal grounding yield subgoals and invariant constraints that guide an augmented MDP with dense rewards for policy optimization.

### 4.1 STL SUBGOAL DECOMPOSITION

Our method of decomposing STL into subgoals with invariant constraints is inspired by Kapoor et al. (2024); Liu et al. (2025). The essence is to first translate the STL into a set of subtasks, where each subtask has a checker $\mu$ on the trace $\sigma$ and belongs to one of the following types:

- **Reachability task**: achieve $\mu(\sigma(\tau)) \geq 0$ at a time instant $\tau$, denoted as $\text{Reach}(\mu, \tau)$.

- **Invariance task**: keep $\mu(\sigma(\tau)) \geq 0$ for all time $\tau$ in an interval $W$, denoted as $\text{Inv}(\mu, W)$.

For basic STL formulas, the time instants and the time intervals can be concrete values or variables: e.g., the formula $G_{[a,b]}\mu$ can be written as $\text{Inv}(\mu, [a,b])$ with concrete $[a,b]$, whereas the formula $F_{[a,b]}\mu$ can be written as $\text{Reach}(\mu, \tau)$ with the time variable $\tau \in [a,b]$, and $\mu_1 U_{[a,b]}\mu_2$ can be written as $\{\text{Reach}(\mu_2, \tau), \text{Inv}(\mu_1, [a,\tau])\}$ with the time variable $\tau \in [a,b]$. For a nested STL, we follow a top-down approach to "flatten" it into reachability and invariance tasks governed by time variables. We denote $\text{Reach}(\phi, \tau)$ for $\rho(\sigma, \tau, \phi) \geq 0$ and use $\text{Inv}(\phi, W)$ to represent $\rho(\sigma, \tau, \phi) \geq 0, \forall \tau \in W$. For any STL $\phi$ we can write it as $\text{Reach}(\phi, 0)$ and then we rewrite with tasks using its subformulas. The subformula will always carry time variables from its ancestor operators, and we repeat the process until all the tasks are represented as atomic propositions (APs) corresponding to $\mu$ or its negation $\neg\mu$. For example, for $\phi = F_{[a,b]}\phi_0 \wedge G_{[c,d]}\neg\mu_0$ where $\phi_0 = \mu_1 \wedge G_{[a_2,b_2]}\mu_2 \wedge F_{[a_3,b_3]}\mu_3$ is a subformula, we can represent $\phi$ as $\{\text{Reach}(\phi_0, \tau), \text{Inv}(\neg\mu_0, [c,d])\}$ with domain $\{\tau \in [a,b]\}$, then we can pass $\tau$ into $\phi_0$ to represent the STL as $\{\text{Reach}(\mu_1, \tau), \text{Inv}(\mu_2, [\tau+a_2, \tau+b_2]), \text{Reach}(\mu_3, \tau+\tau'), \text{Inv}(\neg\mu_0, [c,d])\}$ with domains $\{\tau \in [a,b], \tau' \in [a_3,b_3]\}$. An illustration of the decomposition

---

[1]E.g., if an STL task is to "Eventually reach region A and then reach B", the policy needs to "remember" whether it has already visited the region A in order to proceed to reach B.

is depicted in Fig. 2. In this work, we do not consider disjunctions or temporal structures of the form "$G(F \ldots)$." Such STLs can be represented by introducing additional binary variables to select the disjunction branch and more time variables for each instant in the time domain of the $G$ operator.

From the reachability and invariance tasks, we further denote **subgoals** (reach or stay) as tasks that are either a reachability task (e.g., **Subgoal 1** in Fig. 2) or an invariance task (e.g., **Subgoal 2** in Fig. 2) with atomic proposition $\mu$ (we assume all the APs are for reaching certain regions). The remaining invariance tasks associated with negation of APs (e.g., $\text{Inv}(\neg \mu_0, [c, d])$) are treated as **invariant constraints** (avoidance). Through this decomposition, a complex STL formula $\phi$ reduces to $N_g$ subgoals $\phi_i^g$ with $\text{Reach}(\mu_i^g, \tau_i)$ or $\text{Inv}(\mu_i^g, W_i)$, $i \in \{1, \cdots, N_g\}$ and $N_c$ invariant constraints $\phi_j^c$ with $\text{Inv}(\neg \mu_j^c, W_j)$, $j \in \{1, \cdots, N_c\}$. Denote $\Theta_g$ and $\Theta_c$ as the sets of subgoals and invariant constraints, respectively. Each subgoal / constraint has a starting time and an ending time $[\underline{t}, \overline{t}]$ which is $[\tau, \tau]$ (or $W$). Denote all the time variables in this STL as $\mathbf{t}$. The full algorithm for the temporal decomposition is shown in Algo. 1. Next, we will show how this decomposition guides our state augmentation and reward shaping.

---

**Algorithm 1** Symbolic STL Decomposition

---

1: **Input:** STL formula $\phi$
2: **Output:** Subgoals $\Theta_g$, invariant constraints $\Theta_c$, time variables $\mathbf{t}$
3: **Initialize:** $\Theta_g \leftarrow \emptyset, \Theta_c \leftarrow \emptyset, \mathbf{t} \leftarrow \emptyset$
4: $\text{DECOMPOSE}(\phi, 0)$
5: **function** $\text{DECOMPOSE}(\psi, t_{ref})$
6:     **if** $\psi = \bigwedge_{i=1}^{K} \psi_i$ **then**                                   $\triangleright$ Conjunction operator
7:         **for** $i = 1, \ldots, K$ **do**
8:             $\text{DECOMPOSE}(\psi_i, t_{ref})$
9:         **end for**
10:     **else if** $\psi = \mathbf{G}_{[a,b]} \varphi$ **then**                                $\triangleright$ Always operator
11:         **if** $\varphi$ is of form $\neg\mu$ **then**
12:             $\Theta_c \leftarrow \Theta_c \cup \{\text{Inv}(\neg\mu, [t_{ref} + a, t_{ref} + b])\}$     $\triangleright$ Invariant constraint (avoid)
13:         **else**
14:             $\Theta_g \leftarrow \Theta_g \cup \{\text{Inv}(\mu, [t_{ref} + a, t_{ref} + b])\}$     $\triangleright$ Subgoal (reach-and-stay)
15:         **end if**
16:     **else if** $\psi = \mathbf{F}_{[a,b]} \varphi$ **then**                               $\triangleright$ Eventually operator
17:         Create symbolic variable $\tau \in [a, b]$
18:         $\mathbf{t} \leftarrow \mathbf{t} \cup \{\tau\}$
19:         **if** $\varphi$ is atomic proposition $\mu$ **then**
20:             $\Theta_g \leftarrow \Theta_g \cup \{\text{Reach}(\mu, [t_{ref} + \tau, t_{ref} + \tau])\}$     $\triangleright$ Subgoal (reach)
21:         **else**
22:             $\text{DECOMPOSE}(\varphi, t_{ref} + \tau)$     $\triangleright$ Recursion with updated time
23:         **end if**
24:     **else if** $\psi = \varphi_1 \mathbf{U}_{[a,b]} \varphi_2$ **then**                        $\triangleright$ Until operator
25:         Create symbolic variable $\tau \in [a, b]$
26:         $\mathbf{t} \leftarrow \mathbf{t} \cup \{\tau\}$
27:         **if** $\varphi_2$ is atomic proposition $\mu$ **then**     $\triangleright$ 1. Reachability part (right side)
28:             $\Theta_g \leftarrow \Theta_g \cup \{\text{Reach}(\mu, [t_{ref} + \tau, t_{ref} + \tau])\}$
29:         **else**
30:             $\text{DECOMPOSE}(\varphi_2, t_{ref} + \tau)$
31:         **end if**
32:         **if** $\varphi_1$ is of form $\neg\mu$ **then**     $\triangleright$ 2. Invariance part (left side)
33:             $\Theta_c \leftarrow \Theta_c \cup \{\text{Inv}(\neg\mu, [t_{ref} + a, t_{ref} + \tau])\}$
34:         **else**
35:             $\Theta_g \leftarrow \Theta_g \cup \{\text{Inv}(\mu, [t_{ref} + a, t_{ref} + \tau])\}$
36:         **end if**
37:     **end if**
38: **end function**

---

| Time variables $\mathbf{t} = (\boldsymbol{\tau}, \boldsymbol{\tau}')$ | | | |
|---|---|---|---|
| Task | AP | Starting time $\underline{t}$ | Ending time $\bar{t}$ |
| **Subgoal 1** | $\mu_1$ | $\boldsymbol{\tau}$ | $\boldsymbol{\tau}$ |
| **Subgoal 2** | $\mu_2$ | $\boldsymbol{\tau} + a_2$ | $\boldsymbol{\tau} + b_2$ |
| **Subgoal 3** | $\mu_3$ | $\boldsymbol{\tau} + \boldsymbol{\tau}'$ | $\boldsymbol{\tau} + \boldsymbol{\tau}'$ |
| **Invariant** | $\neg\mu_0$ | $c$ | $d$ |

Figure 2: STL decomposition of $\phi = F_{[a,b]}(\mu_1 \wedge G_{[a_2,b_2]}\mu_2 \wedge F_{[a_3,b_3]}\mu_3) \wedge G_{[c,d]}\neg\mu_0$.

## 4.2 Temporal Grounded State Augmentation and Reward Design

Given a concrete time variables assignment $\mathbf{t}$, the problem is now structured as reaching a sequence of subgoals sorted by their starting time with invariant constraints satisfied during execution. For brevity, we assume the subgoal indices are already sorted. We augment our state as:

$$s = (x, \tau, p_{prev}, p, r, \chi) \tag{3}$$

Here $x \in \mathbb{R}^n$ stands for the original state, $\tau \in \{0, 1, \cdots, T\}$ represents the time index, $p \in \{0, 1, \cdots, N_g\}$ represents the progress index and $p_{prev}$ records the previous progress, $r$ records the certificate to proceed to the next subgoal, $\chi \in \{0, 1\}^{N_c}$ maintains the satisfaction status for the invariant constraints. For the $k$-th subgoal (or invariant constraint), denote the starting time $\underline{t}_k^g$ (or $\underline{t}_k^c$) and the ending time $\bar{t}_k^g$ (or $\bar{t}_k^c$) The augmented state transition can be written as:

$$\begin{cases} x' = f(x,u), \quad \tau' = \tau + 1, \quad p'_{prev} = p, \quad r' = h(r, x', \tau', p'), \quad p' = p + \mathbb{1}(r' = 2) \\ \chi'_k = \chi_k \times \mathbb{1}(\neg(\underline{t}_k^c \leq \tau' \leq \bar{t}_k^c \wedge \neg\mu_k^c(x') < 0)) \quad k = 0, 1, ..., N_c \end{cases} \tag{4}$$

where (in RL implementation, the time index feature input $\tau$ is normalized between 0 and 1):

$$h(r, x', \tau', p') = \begin{cases} 0, & \text{if } r = 2 \\ 1, & \text{if } \underline{t}_{p'}^g \neq \bar{t}_{p'}^g \wedge \tau' = \underline{t}_{p'}^g \wedge \mu_{p'}^g(x') \geq 0 \\ 2, & \text{if } (r = 1 \vee \underline{t}_{p'}^g = \bar{t}_{p'}^g) \wedge (\tau' = \bar{t}_{p'}^g \wedge \mu_{p'}^g(x') \geq 0) \\ r, & \text{otherwise} \end{cases} \tag{5}$$

The variable $r$ acts as a certificate (or flag) that keeps track of whether the reach-and-stay ($FG$) condition has been satisfied. It encodes the progress toward establishing that the predicate holds both at the entry time and the exit time of the required interval. To guide the agent to achieve these subgoals in a proper time window while satisfying the invariant constraints, we design the reward:

$$R(s) = \lambda_1 R_{dist} + \lambda_2 R_{progress} + \lambda_3 R_{success} + \lambda_4 R_{inv} \tag{6}$$

where $R_{dist} = \mu_p^g(x)$ is a distance-based reward shaping to encourage the agent to reach the current subgoal (and stay at the current subgoal within the time window $[\underline{t}_p^g, \bar{t}_p^g]$), $R_{progress} = \mathbb{1}(p^{prev} \neq p)$ encourages the agent to achieve more subgoals, $R_{success} = \mathbb{1}(p = N_g \wedge \chi = \mathbf{1})$ encourages the agent to finish all subgoals without violating any invariant constraints, and $R_{inv} = \mathbb{1}(\chi_k = 0)$ penalizes for violating invariant constraints. The robustness score is also used at the final time step to encourage the agent to satisfy the STL. In this way, the agent is incentivized to reach all the subgoals while obeying the invariant constraints. We use Proximal Policy Optimization (PPO) (Schulman et al., 2017) to train the agent. The policy network and the critic receive the augmented state and the time variable assignment as the input, and output the action and the critic value correspondingly. At the beginning of each training epoch, we sample the time variables and collect episodes to update the network parameters. During inference, we sample time variables and use the trained critic to find the most effective assignment. The most naive way to sample these time variables will be randomly sampling from their feasible intervals, but we will present a better solution in the following section.

## 4.3 Critic-Guided Bayesian Time Allocation

The key challenge in our framework is efficiently searching for time variable assignments. A naive uniform sampling strategy might waste huge effort on assignments that lead to infeasible or low-reward trajectories. To address this, we propose a Bayesian sampling strategy to find promising

time assignments. We do not need to learn an extra surrogate function, as the value function already provides a powerful heuristic. Our prior belief towards "promising" time variables is the uniform distribution. At each epoch, we update our belief using the learned critic value as a non-normalized approximation of the log-likelihood of success, and then we sample the time variables using the Metropolis-Hastings (MH) algorithm. The MH performs a guided random walk over the discrete time variable space and prefers to stay in regions that yield high critic values. To mitigate the risk of the sampler converging to a local optima and the fact that the initial critic might not be accurate, we adopt a hybrid approach: In each epoch, we use an MH sampler to obtain a ratio $\eta_{\text{mcmc}}$ of the time variables and sample a ratio $\eta_{\text{uniform}}$ through uniform sampling. To further leverage knowledge across training epochs, we maintain a replay buffer containing the top $\eta_{\text{elite}}$ ratio of "elite" time variable assignments that yield the highest STL robustness scores. This combination creates a robust and efficient mechanism for discovering effective temporal plans. The full training procedure is detailed in Algo. 2, the MH algorithm is in Algo. 3, and the ablation study is shown in Sec. 5.4.

---

**Algorithm 2** TGPO with Hybrid Time Variable Sampling

---

1: **Input:** STL formula $\phi$ (subgoals and invariant constraints), elite buffer size $K$, batch size $N_B$
2: Initialize policy $\pi_\theta(a|s, \mathbf{t})$, critic $V_\psi(s, \mathbf{t})$, and elite time variable buffer $\mathcal{B}$
3: **for** iteration $i = 1, \ldots, N$ **do**
4:                                                            ▷ *1. High-level Temporal Grounding*
5:     $\mathbf{T}_{\text{uniform}} \leftarrow$ Sample $\eta_{\text{uniform}} N_B$ time variables uniformly from the valid domain $\mathcal{T}$
6:     $\mathbf{T}_{\text{mcmc}} \leftarrow$ Run Metropolis-Hastings guided by $V_\psi$ to generate $\eta_{mcmc} N_B$ time variables.
7:     $\mathbf{T}_{\text{elite}} \leftarrow$ Top $\eta_{elite} N_B$ time variables from elite buffer $\mathcal{B}$
8:     $\mathbf{T}_{\text{batch}} \leftarrow \mathbf{T}_{\text{uniform}} \cup \mathbf{T}_{\text{mcmc}} \cup \mathbf{T}_{\text{elite}}$
9:                                                           ▷ *2. Low-level Policy Optimization*
10:    Collect trajectories $\mathcal{D}_i = \{(\sigma_j, \rho_j^\phi, \mathbf{t}_j)\}$ by executing $\pi_\theta$ with time variables from $\mathbf{T}_{\text{batch}}$
11:    Update $\pi_\theta$ and $V_\psi$ using the PPO algorithm on $\mathcal{D}_i$
12:    Update $\mathcal{B}$ with time variables from $\mathcal{D}_i \cup \mathcal{B}$ corresponding to top-$K$ STL robustness score
13: **end for**
14: **return** Trained policy $\pi_\theta$, critic $V_\psi$, and elite buffer $\mathcal{B}$.

---

## 5 EXPERIMENTS

### 5.1 IMPLEMENTATION DETAILS

**Baselines.** We consider the following approaches. **RNN**: Train RL with a recurrent neural network (RNN) to handle history and use the STL robustness score as the rewards. **CEM**: Cross-Entropy Method (De Boer et al., 2005) that optimizes the policy network with the STL robustness score as the fitness score. **Grad**: A gradient-based method (Meng & Fan, 2023) that trains the policy with a differentiable STL robustness score. $\tau$-**MDP**: An RL method (Aksaray et al., 2016) which augments the state space with a trajectory segment to handle history data. **F-MDP**: An RL approach (Venkataraman et al., 2020) that augments the state space with flags. We denote our base algorithm as **TGPO** and the enhanced version with Bayesian time sampling as **TGPO**[*].

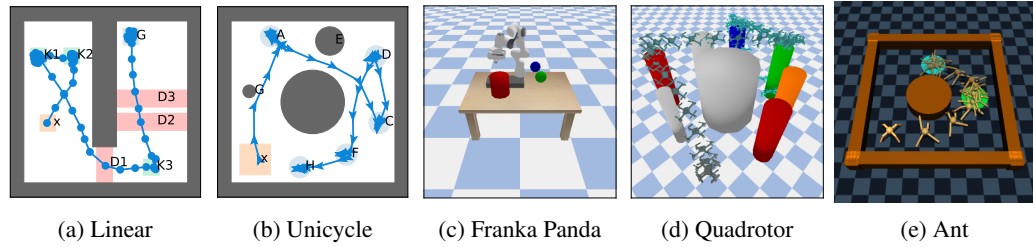

| (a) Linear | (b) Unicycle | (c) Franka Panda | (d) Quadrotor | (e) Ant |

Figure 3: Simulation benchmarks.

**Benchmarks.** We evaluate TGPO across five environments shown in Fig. 3 with varying dynamics and dimensionality: (1) **Linear**: A 2D point-mass linear system. (2) **Unicycle**: A non-holonomic

4D system for a wheeled robot. (3) **Franka Panda**: A 7-DoF robot arm. (4) **Quadrotor**: A 12D, full dynamic model of a quadrotor. (5) **Ant**: a 29D quadruped robot for locomotion tasks. We specify the regions that the agent needs to reach, stay, or avoid and specify their temporal constraints using STL. We designed 10 STL tasks for each benchmark. Five of them are two-layered (e.g., $F_{[0,T]}G_{[0,5]}(\text{Reach A})$), solvable by all the methods. The rest are multi-layer STLs with single-layer or deeper nesting, which cannot be solved by F-MDP. Details can be found in App. A.10.

**Training and evaluation.** For the main comparisons, the task horizon is fixed at $T=100$ except for "Ant" ($T=200$). We trained each model with 7 random seeds to ensure statistical significance. All the methods are implemented in JAX (Bradbury et al., 2018) and trained with 512 parallel environments for 1000~4000 epochs. All experiments were conducted on Amazon Web Services (AWS) g6e.2xlarge instances. A single experiment (a specific set of environment, method, STL, and random seed) took 5 to 90 minutes, depending on the environment and method complexity. In the testing stage, we sample 512 initial states. For each initial state, each baseline is given 10 attempts to generate the solution, and the trajectory with the highest STL robustness score is selected. For our approach, we attempt to select the best time assignment only once, based on the critic value, and then roll out the trajectory (we avoid the use of the STL score as feedback to choose the trajectory). The **Success rate** is the average performance over all the initial states and the STLs. We also measured **Training time**, as shown in App. A.8, which is the time to train each model (averaged over STLs).

## 5.2 MAIN RESULTS

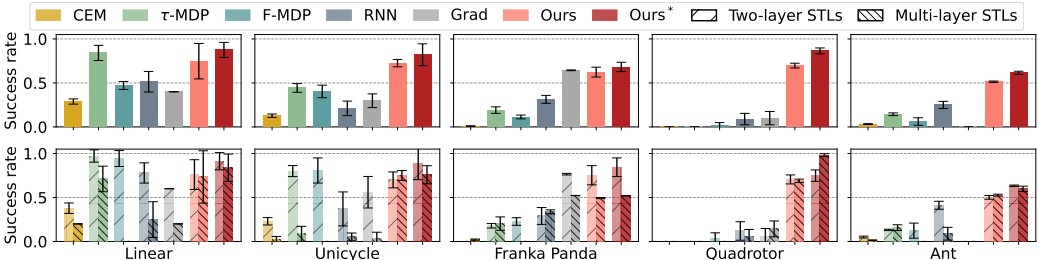

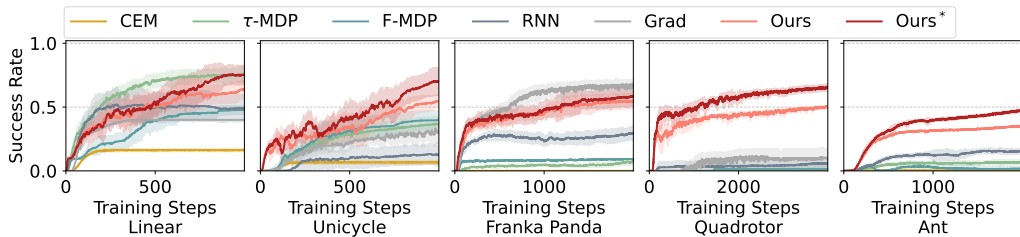

Figure 4: Main comparison. Our method has higher task success rate compared to other baselines.

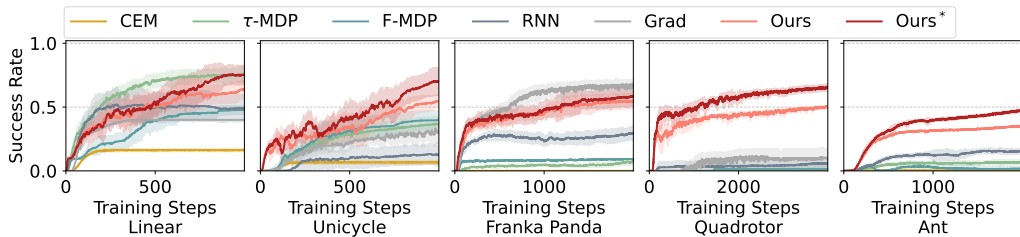

Figure 5: Main comparison for the STL success rate evaluation along the training process.

As shown in Fig. 4 (top row), TGPO achieves the leading performance in most benchmarks, and with Bayesian time variable sampling, TGPO* achieves the highest overall success rate across all benchmarks, indicating the strong empirical performance. Our advantage becomes clearer as the system dimension and the planning difficulty increase, especially in "Quadrotor" and "Ant", where most of the baselines achieve less than 10% success rate, whereas TGPO* can achieve 86.46% and 61.57% success rate, respectively. Under "Linear" system, the best baseline $\tau$-MDP (84.11%) performs competitively compared to TGPO* (87.53%), but $\tau$-MDP's performance drops drastically on the other benchmarks. The "Grad" method is a strong baseline on "Franka Panda", however, its success rate decreases by a large margin on "Quadrotor" due to its complex nonlinear dynamics, and it cannot work at all on "Ant", which is likely caused by the discrepancy between the simulator's approximated gradients and the true non-differentiable dynamics. These findings showcase TGPO's strong performance and great adaptation to high-dimensional and non-differentiable environments. If we look at different types of STLs (Fig. 4, bottom row), on low-dimensional cases ("Linear"

and "Unicycle"), most baselines work well under the simple STL tasks ("two-layer STLs") but they struggle on the harder STLs ("multi-layer STLs", note that F-MDP can only handle "two-layer STLs"). Whereas our approaches (both TGPO and TGPO*) excel at working on these complex STLs and perform consistently well. This shows our approach's strength in handling complex STLs. In Fig. 5, we show the task success rate in training. Our approach can achieve a high task success rate eventually, whereas other baselines show plateauing early in the training.

## 5.3 SOLVING STL WITH DIFFERENT HORIZON-LENGTHS

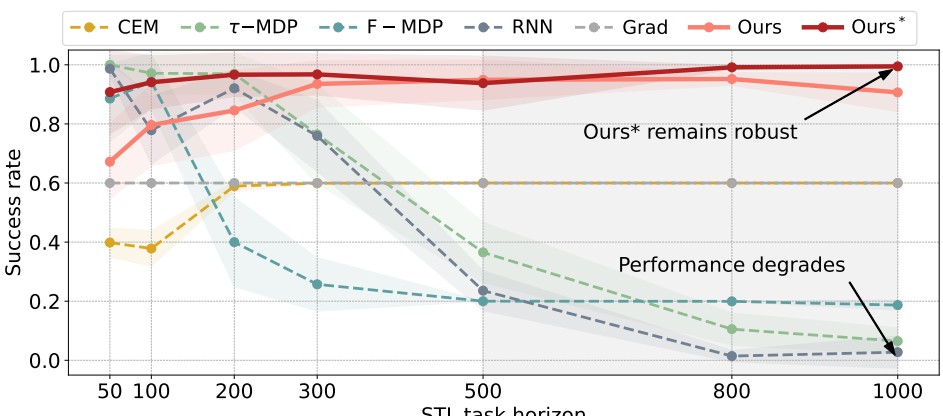

Figure 6: Solving STL in "Linear" environment over varied task horizons. Our method performs the best and maintains high success rate in long horizons where the RL baselines performance degrades.

Beyond system complexity and task difficulty, our methods also show resilient adaptivity for long-horizon tasks. Here, we consider only the two-level STLs and we scale the task horizon to different lengths (50, 200, 300, 800 and 1000). As shown in Fig. 6, our methods (TGPO and TGPO*) keep a high success rate over varied time lengths, whereas for RL methods $\tau$-MDP, F-MDP and RNN, which are strong baselines for shorter horizons ($T$=50 and 100), experience a huge drop in success rate as the horizon increases. It is interesting that CEM and Grad can maintain their performance as the horizon expands 10 times, which may be attributed to their trajectory optimization formulation.

## 5.4 ABLATION STUDIES

Table 1: Ablation studies for TGPO on the linear dynamics environment.

(a) Different time variables sampling strategies.

| Method | Rand. | Bay. | Elite | Test(%) |
|---|---|---|---|---|
| Ours | ✓ | | | $80.33 \pm 8.84$ |
| Ours$_{Bay}$ | | ✓ | | $53.79 \pm 7.99$ |
| Ours$_{Elite}$ | | | ✓ | $61.49 \pm 10.02$ |
| Ours$_{mixBay}$ | ✓ | ✓ | | $81.18 \pm 9.72$ |
| Ours$_{mixElite}$ | ✓ | | ✓ | $86.62 \pm 8.67$ |
| Ours$_{BayElite}$ | | ✓ | ✓ | $81.04 \pm 11.00$ |
| **Ours*** | ✓ | ✓ | ✓ | $\mathbf{88.99 \pm 9.60}$ |

(b) Different state augmentation and rewards.

| State aug. / Reward | Test(%) |
|---|---|
| t+flags / STL | $11.73 \pm 2.67$ |
| t+flags / STL+Inv | $46.85 \pm 10.53$ |
| t+flags / STL+Inv+Prog | $49.80 \pm 7.72$ |
| t+flags / STL+Inv+Dist | $84.59 \pm 7.88$ |
| $\emptyset$ / STL+Inv+Dist | $11.43 \pm 3.48$ |
| t / STL+Inv+Dist | $47.51 \pm 7.86$ |
| **Ours*** (all / all) | $\mathbf{88.99 \pm 9.60}$ |

We conduct a thorough ablation study under "Linear" (all 10 STLs) for the analysis. We first study different sampling strategies. As shown in Tbl. 1a, our base model with random sampling (Ours) can already achieve 80.33% success rate ($\pm$ indicates the standard deviation over 7 random seeds). However, naively using Bayesian sampling (Ours$_{Bay}$) or Elite variable replay buffer (Ours$_{Elite}$) will hurt the performance, likely due to the myopic exploration at the beginning of the training, which restricts the agent from seeking more promising assignments. Hence, we mix the two sources of the

time variables together and witness certain improvement ($\text{Ours}_{\text{mixBay}}$, $\text{Ours}_{\text{mixElite}}$, and $\text{Ours}_{\text{BayElite}}$) compared to Ours. Finally, by combining all these together, $\text{Ours}^*$ achieves the best performance.

In Tbl. 1b we study how state augmentation and reward shaping foster an efficient multi-stage RL. For the reward design, we consider to just using parts of the reward terms introduced before, and the results (the first 4 rows) show that, just using STL robustness score will only result in 11.73% success rate, whereas by gradually adding invariance penalty, progress reward and the distance reward, the performance will get improved (the most improvement comes from using the distance reward term) and finally becomes 88.99% for $\text{Ours}^*$. Regarding the state augmentation, removing the flags in the augmented state will result in a 41.48% drop in success rate, and if further removing the time index counter, the performance will drop to 11.43%. The combined findings validate our design.

## 5.5 VISUALIZATION FOR INTERPRETABILITY AND MULTI-MODAL BEHAVIOR

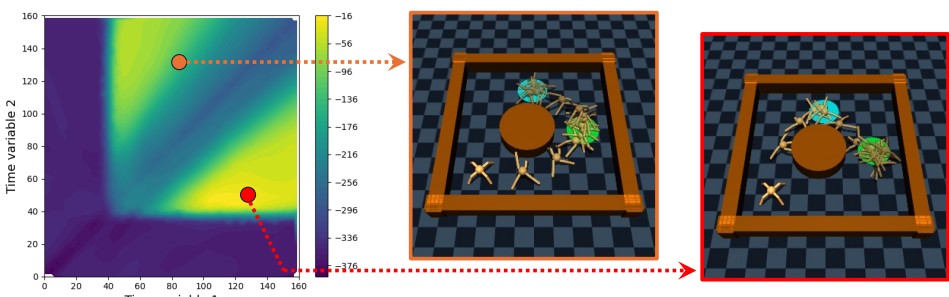

Figure 7: Critic value visualization and simulation for the ant environment under STL $F_{[0,160]}(A_1) \wedge F_{[0,160]}(A_2) \wedge G_{[0,200]} \neg B$. From the two feasible regions on the critic heatmap, we can see that the corresponding conditioned policy generates two behaviors to fulfill the task specification.

TGPO can generate diverse behaviors to fulfill the STL specifications, which can also be reflected from the critic values. We consider an example under the Ant environment for the STL task $F_{[0,160]}(A_1) \wedge F_{[0,160]}(A_2) \wedge G_{[0,200]} \neg B$. The ant starts from the lower left, and there is an obstacle in the middle of the scene. The time variables here correspond to "Reach $A_1$" (the cyan region in the scene) and "Reach $A_2$" (green). After the training, we plot the critic value heatmap across different time variable assignments for the initial state. As shown in Fig. 7, the lower-left L-shape region is in low critic value as it is dynamically infeasible to reach the first subgoal in a short time ($0 \leq \tau \leq 40$). The diagonal line region also receives low critic value, because the two subgoal regions cannot be visited in such a short time. The diagonal line splits the promising time variable regions (yellow) into two parts, from which we can generate two different ways to fulfill the STL task (as shown from the time-elapsed simulation plot on the right). This shows that we can leverage the time variables as the condition to generate multi-modal solutions to solve the STL problem.

## 5.6 LIMITATIONS

While TGPO achieves strong empirical performance, it lacks formal guarantees on convergence to a global optimum. TGPO is effective on STLs with conjunctions and temporal operators, but it might not efficiently handle STLs with disjunctions or infinite-horizon tasks like "Always-Eventually (G(F))". We have tested TGPO with 5 time variables; its scalability towards more complex STLs remains an open question. We have suggested some possible solutions in App. A.11 for future work.

## 6 CONCLUSION

In this paper, we introduce Temporal Grounded Policy Optimization (TGPO), a novel reinforcement learning framework for solving long-horizon Signal Temporal Logic tasks. By using STL decomposition, time variable sampling, state augmentation and reward design, TGPO can effectively handle complex STL and dynamics for long-horizon tasks. Experiments show that TGPO significantly outperforms existing baselines across various robotic environments and STL formulas. Future work will focus on extending TGPO to handle a broader class of STL formulas and improving its scalability.

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

# A APPENDIX

## A.1 THE USE OF LARGE LANGUAGE MODELS (LLMS)

In this submission, we used LLM to help with refining the manuscript's prose for clarity and generating plots based on our experimental results. All novel hypotheses, methodological innovations, experimental results, and scientific conclusions presented in this manuscript are the exclusive intellectual work of the authors. In accordance with ICLR policy, we have thoroughly verified and edited all LLM-assisted content and bear full responsibility for the final manuscript.

## A.2 ALGORITHM HYPERPARAMETERS

All the main hyperparameters used during training are shown in Table 2.

## A.3 SIMULATION ENVIRONMENT DETAILS

In this paper, we conduct experiments on five simulation environments (Linear, Unicycle, Franka Panda, Quadrotor, and Ant). The first four environments were implemented in plain JAX code by writing out the system dynamics, whereas the last one was adopted from the Mujoco JAX implementation. Detailed implementations are listed as follows.

Table 2: Hyperparameters assignments used for training TGPO[*].

| Hyperparameter | Linear; Unicycle; FrankaPanda; Quadrotor; Ant |
|---|---|
| Network hidden units | (512, 512, 512) |
| Optimizer | Adam |
| Learning rate | $3 \times 10^{-4}$ |
| Weight decay | 0.1 |
| Grad norm clip | 0.5 |
| Random seeds | 1007,1008,1009,1010,1011,1012,1013 |
| Batch size | 512 |
| Epochs | 1000 (L, U); 2000 (F, A); 4000(Q) |
| Time steps $T$ | 100 (L, U, F, Q); 200 (A) |
| Time duration $\Delta t$ | 0.2 (L, U); 0.05 (F, A); 0.1 (Q) |
| Distance reward $\lambda_1$ | 0.5 |
| Progress reward $\lambda_2$ | 20.0 |
| Success reward $\lambda_3$ | 20.0 |
| Invariance penalty $\lambda_4$ | -3.0 (L, F, Q); -3.5 (U); -1.5 (A) |
| Number of MCMC steps $N_{MCMC}$ | 500 |
| Number of warmup steps $N_{warmup}$ | 200 |
| Number of MCMC chains $M_{MCMC}$ | 512 |
| Ratio of Randomly-sampled time variables $\eta_{mcmc}$ | 0.5 |
| Ratio of MCMC-sampled time variables $\eta_{uniform}$ | 0.4 |
| Ratio of Elite time variables $\eta_{elite}$ | 0.1 |
| Elite buffer size $|\mathcal{B}|$ | 512 |

### A.3.1 LINEAR

We use a single-integrator dynamics model. The 2D state $(x, y)^T$ represents the 2D coordinates on a xy-plane, and the 2D control input $(v, w)^T$ reflects the velocities in these two directions. The system dynamics is described as:

$$\begin{cases} x_{t+1} = x_t + v_t \Delta t \\ y_{t+1} = y_t + w_t \Delta t \end{cases} \tag{7}$$

We set the time step duration $\Delta t = 0.2s$.

### A.3.2 UNICYCLE

We use a car-like dynamics model. The 4D state $(x, y, \theta, v)^T$ represents the 2D coordinates on the xy-plane, the heading angle of the robot and the velocity of the robot, respectively. The 2D input $(\omega, a)^T$ represents the angular velocity and the acceleration. The system dynamics can be described as:

$$\begin{cases} x_{t+1} = x_t + v_t \cos(\theta_t) \Delta t \\ y_{t+1} = y_t + v_t \sin(\theta_t) \Delta t \\ \theta_{t+1} = \theta_t + \omega_t \Delta t \\ v_{t+1} = v_t + a_t \Delta t \end{cases} \tag{8}$$

We set the time step duration $\Delta t = 0.2s$. The control actuation is limited at $[-1rad/s, +1rad/s] \times [-4m/s^2, +4m/s^2]$. The scene layout is $[-5m, +5m] \times [-5m, +5m]$ on the xy-plane.

### A.3.3 FRANKA PANDA

We use a 7 DoF Franka Panda robot arm model to conduct the simulation. The 7D state $(\theta_1, \theta_2, ..., \theta_7)^T$ represents the angle for all the joints where $\theta_7$ is for the end-effector joint. The 7D control input $(\omega_1, \omega_2, ..., \omega_7)^T$ represents the angular velocity for all the joints. The dynamics follows a simple single-integrator case: $\theta_{i,t+1} = \theta_{i,t} + \omega_{i,t} \Delta t$, for $i = 1, 2, ..., 7$. We set the time step duration $\Delta t = 0.05s$.

### A.3.4 QUADROTOR

We use a full quadrotor dynamics model (Tayebi & McGilvray, 2006) to conduct the simulation. The 12D state $(x, y, z, v_x, v_y, v_z, \phi, \theta, \psi, \omega_x, \omega_y, \omega_z)^T$ represents the 3D coordinate $\mathbf{p} = (x, y, z)^T$, the velocity vector $\mathbf{v} = (v_x, v_y, v_z)^T$, the orientation vector $\boldsymbol{\Theta} = (\phi, \theta, \psi)^T$, and the angular velocity $\boldsymbol{\omega} = (\omega_x, \omega_y, \omega_z)^T$, respectively. The 4D control input $(f_1, f_2, f_3, f_4)^T$ represents the lifting force from the four motors. The full dynamics are:

$$
\begin{cases}
\mathbf{p}_{t+1} = \mathbf{p}_t + \mathbf{v}_t \Delta t \\
\mathbf{v}_{t+1} = \mathbf{v}_t + (g\mathbf{e}_3 - \frac{T}{m} R_z(\psi) R_y(\theta) R_x(\phi) \mathbf{e}_3) \Delta t \\
\boldsymbol{\Theta}_{t+1} = \boldsymbol{\Theta}_t + \boldsymbol{\omega}_t \Delta t \\
\boldsymbol{\omega}_{t+1} = \boldsymbol{\omega}_t + I^{-1}(\boldsymbol{\tau} - \boldsymbol{\omega}_t \times (I\boldsymbol{\omega}_t)) \Delta t
\end{cases}
\tag{9}
$$

with the rotation matrices $R_x(\phi) = \begin{bmatrix} 1 & 0 & 0 \\ 0 & \cos(\phi) & -\sin(\phi) \\ 0 & \sin(\phi) & \cos(\phi) \end{bmatrix}$, $R_y(\theta) = \begin{bmatrix} \cos(\theta) & 0 & \sin(\theta) \\ 0 & 1 & 0 \\ -\sin(\theta) & 0 & \cos(\theta) \end{bmatrix}$, and $R_z(\psi) = \begin{bmatrix} \cos(\psi) & -\sin(\psi) & 0 \\ \sin(\psi) & \cos(\psi) & 0 \\ 0 & 0 & 1 \end{bmatrix}$ and $T$ and $\boldsymbol{\tau}$ are the total thrust and the torques derived from the motor input $u$ with the Coriolis effect considered to the angular velocity vector. We set the time step duration $\Delta t = 0.10s$, adapt the gravity coefficient $g = 9.81m/s^2$ with the corresponding gravity vector $\mathbf{e}_3 = (0, 0, 1)^T$, set the total mass of the quadrotor $m = 0.2kg$ and set the diagonal line of the quadrotor inertia matrix $I$ as $(0.01kg \cdot m^2, 0.01kg \cdot m^2, 0.02kg \cdot m^2)^T$.

### A.3.5 ANT

In this case, the agent is a 8-DoF quadruped robot with the complex dynamics implemented in Brax (Freeman et al., 2021). The observation space is 29-dimension (3-dimension for xyz coordinates, 4-dimension for the torso orientation (in Quaternion representation), 3-dimension velocity vector and 3-dimension angular velocity for the torso, 8-dimension for the joints' angles and another 8-dimension for the joints' angular velocities). The original control input is 8-dimension for the torques applied to each of the 8 joints. To ease the RL training, we first train a goal-reaching policy, enabling the ant to learn and move to a specified target location. Then, for the baselines and our methods, the problem becomes planning the waypoints so that the ant can satisfy the STL tasks specified.

### A.4 BASELINE IMPLEMENTATION DETAILS

### A.4.1 CEM

We use the Cross Entropy Method baseline mentioned in (Meng & Fan, 2023), which belongs to the evolutionary search algorithm mentioned in (Salimans et al., 2017). We denote the initial neural network policy parameters as $\theta^{(0)}$. At $j$-th iteration, we draw $N$ samples $\theta_1, ... \theta_N$ from $\mathcal{N}(\theta^{(j)}, \sigma^{(j)^2})$ where $\sigma^{(j)}$ is the preset standard deviation, then we rollout the trajectories and compute their robustness score. We pick the top-$K$ candidates parameters $\theta_{E_1}, ... \theta_{E_k}$. Then we update the estimate for the neural network parameters $\theta^{(j+1)} = \frac{1}{k} \sum_{i=1}^{k} \theta_{E_i}$ and $\sigma^{(j+1)} = \sqrt{\frac{1}{k-1} \sum_{i=1}^{k} (\theta_{E_i} - \theta^{(j+1)})^2}$. We repeat this process for $L$ iterations to get the final parameters. We set the size for the elite pool to be $K = 32$ and set the population sample size to be $N = 512$. The number of iteration steps $L$ is the same as our method ($L = 1000$ for "Linear" and "Unicycle", $L = 2000$ for "Franka Panda" and "Ant", and $L = 4000$ for "Quadrotor".)

### A.4.2 $\tau$-MDP

$\tau$-MDP is an RL method introduced in (Aksaray et al., 2016) to solve STL tasks under the discrete state space. The original method appends history to the state space, and uses Q-learning to solve short-horizon tasks with 2-layer STL specifications. Here, we extend it to handle general STL

formulas by augmenting the entire trajectory into the state space with STL robustness score as the terminal reward to guide the agent to satisfy STL tasks. We also changed the RL backbone from Q-learning to PPO for better scalability to longer-horizon tasks (The original Q-learning tabular formulation will not work on continuous space for $T = 100$).

### A.4.3  $F$-MDP

$F$-MDP is an improved RL method introduced in (Venkataraman et al., 2020) to solve STL tasks under the discrete state space more efficiently. This approach considers the 2-layer STL specifications, and introduces a flag for each of the subformulas in the STL. They defined the state transition rules and reward mechanism for "F" and "G"-based subformulas based on these flags and show that the Q-learning under this augmentation can learn more efficiently than the Q-learning under $\tau$-MDP (Aksaray et al., 2016). We re-implemented $F$-MDP in PPO for our comparison.

### A.4.4  RNN

In this case, similar to (Liu et al., 2021), we use an RNN to encode the history data and then use the robustness score as the final reward to guide the agent to satisfy the tasks. The issue of this implementation is that it is much more time-consuming compared to the other baselines.

### A.4.5  GRAD

In this case, similar to (Leung & Pavone, 2022) and (Meng & Fan, 2023), we use a neural network policy to roll out the trajectory (in a deterministic manner, rather than sampling from the learned Gaussian distribution). At each time step, the network receives the state (and the time index) and generates the action, which is then sent to the environment to derive the next state. We repeat this process $T$ times to roll out the full trajectory, which preserves the gradient through the differentiable system dynamics. We use the approximated robustness score mentioned in (Pant et al., 2017) to ensure the score is differentiable. We then conduct backpropagation-through-time (BPTT) to update the neural network parameters.

### A.5  THEORETICAL ANALYSIS OF REWARD CONSISTENCY TOWARDS STL SATISFACTION

Here we show that under the reward dominant assumption, the optimal policy that maximizes the expected return is able to fulfill the task specification.

**Theorem 1 (Consistency of reward design and STL satisfaction)** *Assume the problem is feasible and consider a fixed initial state $s_0$. Let $Z(\tau) = \sum_{t=0}^{T} \gamma^t \mu(s_t)$ be the cumulative discounted "$R_{dist}$"-related return of the trajectory $\tau$ (we omit the subscripts for $\mu$ for brevity). Let $\Delta Z = \max_{\tau} Z(\tau) - \min_{\tau'} Z(\tau')$ be the maximum difference in $Z$ between any two trajectories. If $\exists \lambda_1, \lambda_2, \lambda_3, \lambda_4$ such that:*

$$\gamma^T(N_g \lambda_2 + \lambda_3) - (N_g - 1)\lambda_2 > \lambda_1 \Delta Z,$$

$$\gamma^T(\lambda_3 - \lambda_4) - N_g \lambda_2(1 - \gamma^T) > \lambda_1 \Delta Z,$$

*then the optimal policy of TGPO will satisfy the STL task.*

**Proof 1** *To ensure the optimal policy satisfies the STL, the worst-case discounted return from the satisfiable trajectories should be higher than the best-case discounted return from non-satisfiable trajectories.*

*1. The return for the satisfiable trajectories $G_{sat} \geq \lambda_1 \min_{\tau} Z(\tau) + \lambda_2 N_g \gamma^T + \lambda_3 \gamma^T$, where the first part accounts for the reward-shaping, the second part is for progress reward (the worst-case will be all the goals are reached at very last steps), and the last is for the task success.*

*For the non-satisfiable trajectories, there are two possible cases: (1) failure due to not reaching all the goals, or failure due to violating the invariant properties.*

*2. The return for non-reaching-all-goal trajectories $G_{inc} \leq \lambda_1 \max_{\tau} Z(\tau) + \lambda_2(N_g - 1)$. (The highest return is achieved by visiting all but the last subgoals in the earliest possible time, and satisfying all the invariant constraints.)*

3. *The return for invariance-violation trajectories $G_{vio} \leq \lambda_1 \max_\tau Z(\tau) + \lambda_2 N_g + \lambda_4 \gamma^T$. (The highest return is achieved by visiting all the subgoals in the earliest possible time, and just violating one constraint in the last time step.)*

*To ensure the optimal policy solves the STL task, we should have $G_{inc} \geq \max(G_{inc}, G_{vio})$, thus:*

$$\lambda_1 \min_\tau Z(\tau) + \lambda_2 N_g \gamma^T + \lambda_3 \gamma^T > \lambda_1 \max_\tau Z(\tau) + \lambda_2 (N_g - 1),$$

$$\lambda_1 \min_\tau Z(\tau) + \lambda_2 N_g \gamma^T + \lambda_3 \gamma^T > \lambda_1 \max_\tau Z(\tau) + \lambda_2 N_g + \lambda_4 \gamma^T.$$

*Therefore, we have*

$$\gamma^T (N_g \lambda_2 + \lambda_3) - (N_g - 1)\lambda_2 > \Delta Z,$$

$$\gamma^T (\lambda_3 - \lambda_4) - N_g \lambda_2 (1 - \gamma^T) > \Delta Z.$$

### A.6 SOLVING STL WITH VARIED HORIZON LENGTHS FOR THE UNICYCLE ENVIRONMENT.

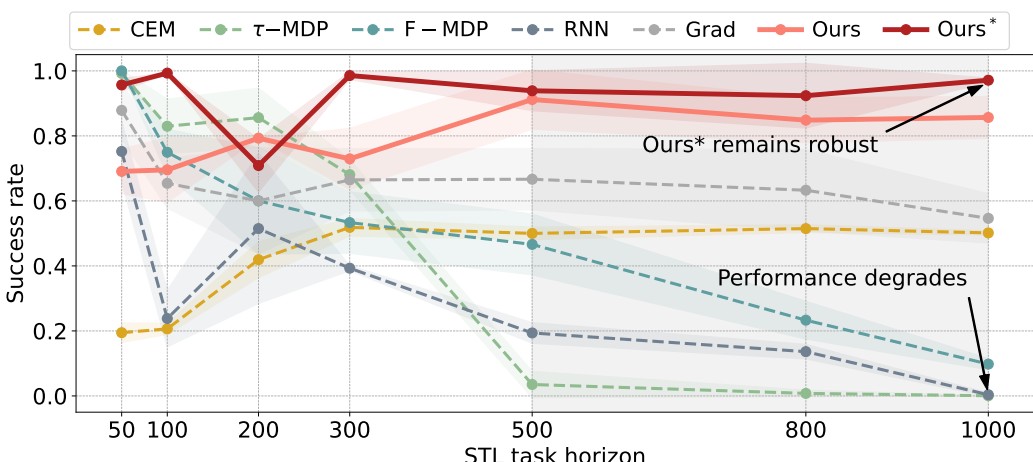

Figure 8: Solving STL in "Unicycle" environment over varied task horizons. Here we train each model with 3 random seeds. Similar to the trend in Fig. 6, our methods (TGPO and TGPO*) have the leading performance in $T$=100 case, and maintain high success rate in long horizons where RL baselines performance degrades.

We conduct experiments on the nonlinear "unicycle" environment to compare performance under varied task horizons (similar to Sec. 5.3, the temporal operator intervals are scaled accordingly). As shown in Fig. 8, our approaches (TGPO and TGPO*) have the competitive performance for short horizon settings ($T$=50,100), and maintain high success rate in longer horizons ($T$=300,500,800,1000) whereas other RL baselines encounter a degradation in the task success rate. The trend is similar to the one conducted under the "linear" environment (shown in Fig. 6), which showcases our approach's effectiveness in handling long-horizon tasks is consistent across different system dynamics.

### A.7 TEMPORAL SAMPLING ALGORITHM DETAILS

The Metropolis-Hastings algorithm (Chib & Greenberg, 1995) is a Markov Chain Monte Carlo (MCMC) method for sampling from a probability distribution, commonly used when directly sampling from the distribution is hard. In our approach TGPO*, we use a discrete version of the M-H algorithm to sample time variables $\mathbf{t}$ that are likely to yield high critic values $V_\psi(s_0, \mathbf{t})$, where $s_0$ is the initial state. We use $\exp(V_\psi(s_0, \mathbf{t}))$ as a proxy for the unnormalized probability of the promising temporal variables. The algorithm proceeds by starting with an initial set of temporal variables $\mathbf{t}_0$ and iteratively proposing to move on grids to a new set $\mathbf{t}'$ based on a proposal distribution $g(\mathbf{t}'|\mathbf{t})$. The move is then accepted or rejected based on the acceptance ratio $\alpha$, which compares the critic value exponentials of the new and the current variables. The process is detailed in Algorithm. 3.

---

**Algorithm 3** Metropolis-Hastings for time variable sampling (with multiple chains and warm-up)

---

1: **Input:** Initial state $s_0$, Critic network $V_\psi(s, \mathbf{t})$, Proposal distribution $g(\mathbf{t}'|\mathbf{t})$
2: **Input:** Iterations $N_{mcmc}$, Number of chains $M_{chain}$, Number of warm-up steps $N_{warmup}$
3: **for all** chain $m \in \{1, \ldots, M_{chain}\}$ **do**
4:      Initialize temporal variables $\mathbf{t}_{m,0}$ randomly
5:      Initialize samples list $S_m \leftarrow []$
6: **end for**
7: **for** $i = 1$ **to** $N_{mcmc}$ **do**
8:      **for all** chain $m \in \{1, \ldots, M_{chain}\}$ **do**
9:          ▷ Propose new temporal variables for chain $m$
10:          $\mathbf{t}' \leftarrow$ Sample from $g(\mathbf{t}'|\mathbf{t}_{m,i-1})$
11:          ▷ Calculate the acceptance ratio $\alpha$
12:          $Q_{\text{current}} \leftarrow V_\psi(s_0, \mathbf{t}_{m,i-1})$
13:          $Q_{\text{new}} \leftarrow V_\psi(s_0, \mathbf{t}')$
14:          $\alpha \leftarrow \min(1, \exp(Q_{\text{new}} - Q_{\text{current}}))$
15:          ▷ Accept or reject the new sample for chain $m$
16:          $u \leftarrow$ Sample from $\text{Uniform}(0, 1)$
17:          **if** $u < \alpha$ **then**
18:              $\mathbf{t}_{m,i} \leftarrow \mathbf{t}'$          ▷ Accept the new sample
19:          **else**
20:              $\mathbf{t}_{m,i} \leftarrow \mathbf{t}_{m,i-1}$          ▷ Reject and keep the old sample
21:          **end if**
22:      **end for**
23: **end for**
24:          ▷ Collect samples after the warm-up period
25: **for** $i = N_{warmup} + 1$ **to** $N_{mcmc}$ **do**
26:      **for all** chain $m \in \{1, \ldots, M_{chain}\}$ **do**
27:          Add $\mathbf{t}_{m,i}$ to $S_m$
28:      **end for**
29: **end for**
30: **Return:** Sampled time variables $\{\mathbf{t}|\mathbf{t} \in S_i, i = 1, 2, \ldots, M\}$

---

In our approach, we set $N_{mcmc} = 500$, $N_{warmup} = 200$, $M_{chain} = 512$ and pick the time variable from each $S_i$ with the highest critic value to form the time variable set $\mathbf{T}_{mcmc}$ used in Alg. 2. For the proposal distribution $g(\mathbf{t}'|\mathbf{t})$, we use a uniform distribution over the local neighborhood of the current temporal variables $\mathbf{t}$: we first uniformly sample an index $j$ from the dimensions of $\mathbf{t}$ and then uniformly sample a move direction $\Delta \in \{-1, +1\}$. The proposed new set of variables $\mathbf{t}'$ is generated by applying this change to the selected index but also ensure that the new value $\mathbf{t}'$ is within the valid range (otherwise we keep $\mathbf{t}$ unchanged).

### A.8   Training time comparison

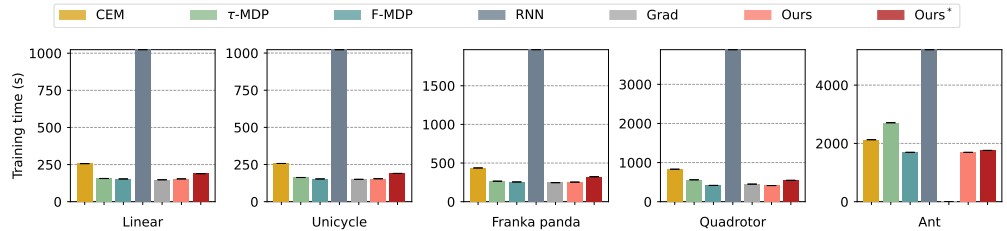

Figure 9: Comparison for training time. For each method under each environment, the result is averaged over 10 STLs and 7 random seeds. TGPO's training time is on par with leading baselines.

As shown in Fig. 9, TGPO and TGPO[*] have a similar runtime compared to $\tau$-MDP, $F$-MDP and Grad baselines, whereas the CEM baseline is normally 20.8%~35.8% higher than TGPO[*]. The most time-consuming baseline is RNN, where TGPO[*] is 1.96X~6.11X faster in training speed. This

shows that our approach is as scalable as other top RL baselines in training time, but our method can achieve higher task success rate.

## A.9 CORRELATION BETWEEN THE CRITIC AND THE STL ROBUSTNESS SCORE

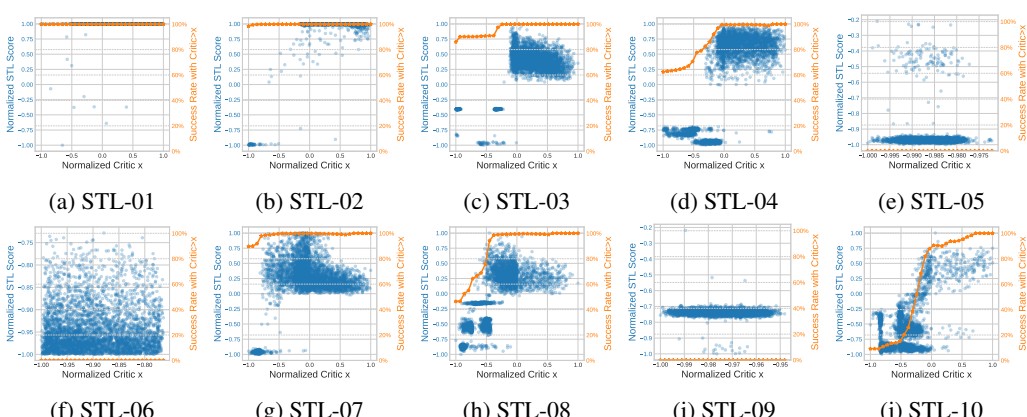

| (a) STL-01 | (b) STL-02 | (c) STL-03 | (d) STL-04 | (e) STL-05 |
| (f) STL-06 | (g) STL-07 | (h) STL-08 | (i) STL-09 | (j) STL-10 |

Figure 10: Correlation analysis between the critic and the STL score (seed=1007).

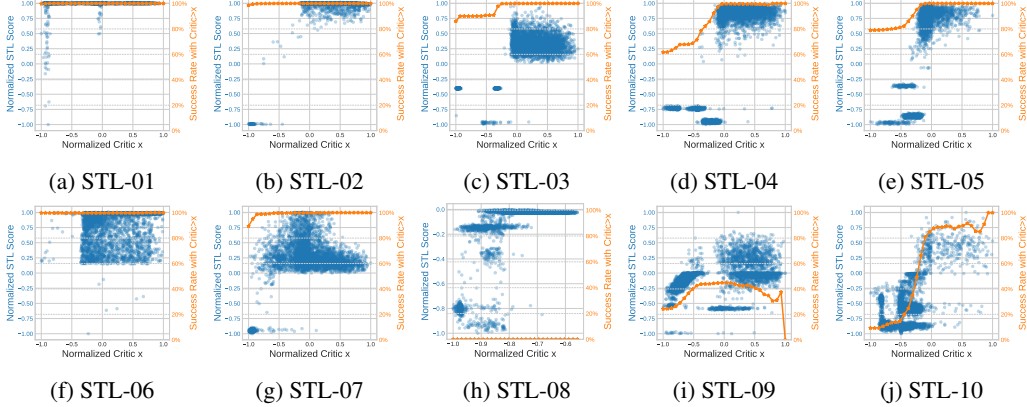

| (a) STL-01 | (b) STL-02 | (c) STL-03 | (d) STL-04 | (e) STL-05 |
| (f) STL-06 | (g) STL-07 | (h) STL-08 | (i) STL-09 | (j) STL-10 |

Figure 11: Correlation analysis between the critic and the STL score (seed=1008).

To validate that our learned critic in TGPO can really reflect the "promising" time variables that lead to STL satisfaction, in the "Linear" environment, for the TGPO algorithm, we randomly sample 4096 points from the pretrained critic and rollout the corresponding trajectories to generate the STL robustness score. We plot the (critic value, STL score) scatter plot, together with the cumulative STL success rate curve for samples with a critic value greater than x. As shown in Fig. 10 (for seed=1007) and Fig. 11 (for seed=1008) from the blue scatter plots, whenever the critic value (left) is higher, the STL score is more likely to be higher, and hence more likely to satisfy STL. If we look at the orange curve, as the Critic value x increases, in most cases the probability for the corresponding traces satisfying the STL score is monotonously increasing or plateau at 100%, which indicates that our critic is learned correctly (note that if the critic is not learned well, it could learn for some time variables that bring in high critic value but result in low STL scores, like STL-09 in Fig. 11) and can be used to find "promising" time variable assignments.

## A.10 STL TASK DETAILS

Under each simulation environment, we make 10 STL formulas in two different categories ("two-level" and "multi-level"). Here we only consider predicates related with "Reach", "Stay", "Avoid" certain objects in the scene. They are listed as follows.

### A.10.1 STLs in "Linear" environment

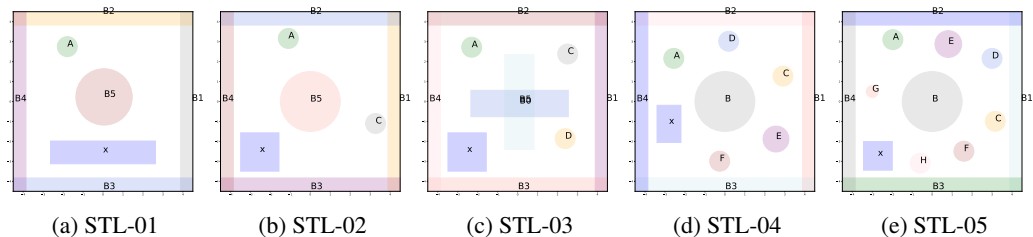

(a) STL-01  (b) STL-02  (c) STL-03  (d) STL-04  (e) STL-05

Figure 12: Scene for Linear: STL tasks 01 to 05

**STL-01 (Two-layer):** $F_{[5:7]}(F_{[50:85]}(A) \wedge G_{[0:90]}(\neg B_5) \wedge G_{[0:90]}(\neg B_1) \wedge G_{[0:90]}(\neg B_2) \wedge G_{[0:90]}(\neg B_3) \wedge G_{[0:90]}(\neg B_4))$

**STL-02 (Two-layer):** $F_{[5:10]}(F_{[0:50]}(A) \wedge G_{[60:80]}(C) \wedge G_{[0:90]}(\neg B_5) \wedge G_{[0:90]}(\neg B_1) \wedge G_{[0:90]}(\neg B_2) \wedge G_{[0:90]}(\neg B_3) \wedge G_{[0:90]}(\neg B_4))$

**STL-03 (Two-layer):** $F_{[5:10]}(F_{[0:50]}(A) \wedge F_{[40:60]}(C) \wedge G_{[70:80]}(D) \wedge G_{[0:90]}(\neg B_5) \wedge G_{[0:90]}(\neg B_0) \wedge G_{[0:90]}(\neg B_1) \wedge G_{[0:90]}(\neg B_2) \wedge G_{[0:90]}(\neg B_3) \wedge G_{[0:90]}(\neg B_4))$

**STL-04 (Two-layer):** $F_{[5:10]}(F_{[0:50]}(A) \wedge F_{[40:50]}(C) \wedge F_{[70:80]}(F) \wedge G_{[50:60]}(D) \wedge G_{[0:90]}(\neg B) \wedge G_{[0:90]}(\neg E) \wedge G_{[0:90]}(\neg B_1) \wedge G_{[0:90]}(\neg B_2) \wedge G_{[0:90]}(\neg B_3) \wedge G_{[0:90]}(\neg B_4))$

**STL-05 (Two-layer):** $F_{[5:10]}(F_{[0:30]}(A) \wedge F_{[30:50]}(C) \wedge F_{[70:80]}(F) \wedge F_{[75:88]}(H) \wedge G_{[50:60]}(D) \wedge G_{[0:90]}(\neg B) \wedge G_{[0:90]}(\neg E) \wedge G_{[0:90]}(\neg G) \wedge G_{[0:90]}(\neg B_1) \wedge G_{[0:90]}(\neg B_2) \wedge G_{[0:90]}(\neg B_3) \wedge G_{[0:90]}(\neg B_4))$

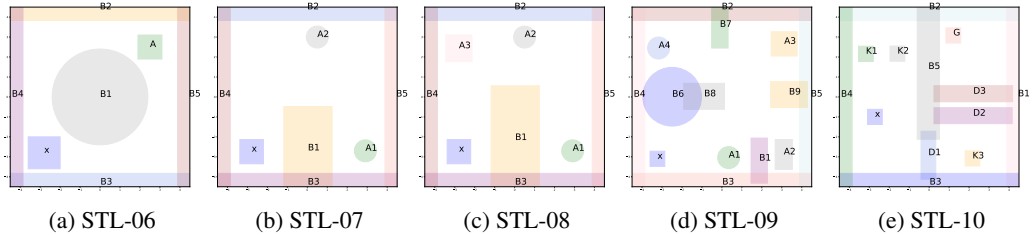

(a) STL-06  (b) STL-07  (c) STL-08  (d) STL-09  (e) STL-10

Figure 13: Scene for Linear: STL tasks 06 to 10

**STL-06 (Multi-layer):** $F_{[10:90]}(A) \wedge G_{[0:100]}(\neg B_1) \wedge G_{[0:100]}(\neg B_2) \wedge G_{[0:100]}(\neg B_3) \wedge G_{[0:100]}(\neg B_4) \wedge G_{[0:100]}(\neg B_5)$

**STL-07 (Multi-layer):** $F_{[0:90]}(A_1) \wedge F_{[40:80]}(A_2) \wedge G_{[0:100]}(\neg B_1) \wedge G_{[0:100]}(\neg B_2) \wedge G_{[0:100]}(\neg B_3) \wedge G_{[0:100]}(\neg B_4) \wedge G_{[0:100]}(\neg B_5)$

**STL-08 (Multi-layer):** $F_{[0:90]}(A_1) \wedge F_{[40:80]}(A_2 \wedge F_{[10:20]}(G_{[0:10]}(A_3))) \wedge G_{[0:100]}(\neg B_1) \wedge G_{[0:100]}(\neg B_2) \wedge G_{[0:100]}(\neg B_3) \wedge G_{[0:100]}(\neg B_4) \wedge G_{[0:100]}(\neg B_5)$

**STL-09 (Multi-layer):** $F_{[5:20]}(A_1 \wedge F_{[10:20]}(G_{[0:5]}(A_2) \wedge F_{[10:30]}(G_{[0:5]}(A_3)) \wedge F_{[10:30]}(G_{[0:10]}(A_4)))) \wedge G_{[0:100]}(\neg B_1) \wedge G_{[0:100]}(\neg B_2) \wedge G_{[0:100]}(\neg B_3) \wedge G_{[0:100]}(\neg B_4) \wedge G_{[0:100]}(\neg B_5) \wedge G_{[0:100]}(\neg B_6) \wedge G_{[0:100]}(\neg B_7) \wedge G_{[0:100]}(\neg B_8) \wedge G_{[0:100]}(\neg B_9)$

**STL-10 (Multi-layer):** $(\neg D_1)U_{[0:100]}(K_1) \wedge (\neg D_2)U_{[0:100]}(K_2) \wedge (\neg D_3)U_{[0:100]}(K_3) \wedge F_{[80:90]}(G_{[0:5]}(G)) \wedge G_{[0:100]}(\neg B_1) \wedge G_{[0:100]}(\neg B_2) \wedge G_{[0:100]}(\neg B_3) \wedge G_{[0:100]}(\neg B_4) \wedge G_{[0:100]}(\neg B_5)$

### A.10.2 STLs in "Unicycle" environment

**STL-01 (Two-layer):** $F_{[5:7]}(F_{[50:85]}(A) \wedge G_{[0:90]}(\neg B_5) \wedge G_{[0:90]}(\neg B_1) \wedge G_{[0:90]}(\neg B_2) \wedge G_{[0:90]}(\neg B_3) \wedge G_{[0:90]}(\neg B_4))$

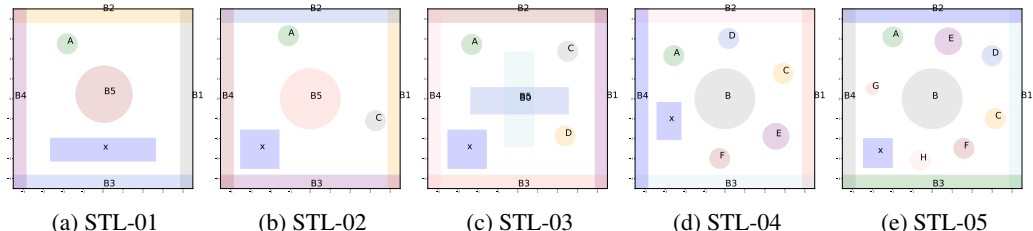

Figure 14: Scene for Unicycle: STL tasks 01 to 05

**STL-02 (Two-layer):** $F_{[5:10]}(F_{[0:50]}(A) \wedge G_{[60:80]}(C) \wedge G_{[0:90]}(\neg B_5) \wedge G_{[0:90]}(\neg B_1) \wedge G_{[0:90]}(\neg B_2) \wedge G_{[0:90]}(\neg B_3) \wedge G_{[0:90]}(\neg B_4))$

**STL-03 (Two-layer):** $F_{[5:10]}(F_{[0:50]}(A) \wedge F_{[40:60]}(C) \wedge G_{[70:80]}(D) \wedge G_{[0:90]}(\neg B_5) \wedge G_{[0:90]}(\neg B_0) \wedge G_{[0:90]}(\neg B_1) \wedge G_{[0:90]}(\neg B_2) \wedge G_{[0:90]}(\neg B_3) \wedge G_{[0:90]}(\neg B_4))$

**STL-04 (Two-layer):** $F_{[5:10]}(F_{[0:50]}(A) \wedge F_{[40:50]}(C) \wedge F_{[70:80]}(F) \wedge G_{[50:60]}(D) \wedge G_{[0:90]}(\neg B) \wedge G_{[0:90]}(\neg E) \wedge G_{[0:90]}(\neg B_1) \wedge G_{[0:90]}(\neg B_2) \wedge G_{[0:90]}(\neg B_3) \wedge G_{[0:90]}(\neg B_4))$

**STL-05 (Two-layer):** $F_{[5:10]}(F_{[0:30]}(A) \wedge F_{[30:50]}(C) \wedge F_{[70:80]}(F) \wedge F_{[75:88]}(H) \wedge G_{[50:60]}(D) \wedge G_{[0:90]}(\neg B) \wedge G_{[0:90]}(\neg E) \wedge G_{[0:90]}(\neg G) \wedge G_{[0:90]}(\neg B_1) \wedge G_{[0:90]}(\neg B_2) \wedge G_{[0:90]}(\neg B_3) \wedge G_{[0:90]}(\neg B_4))$

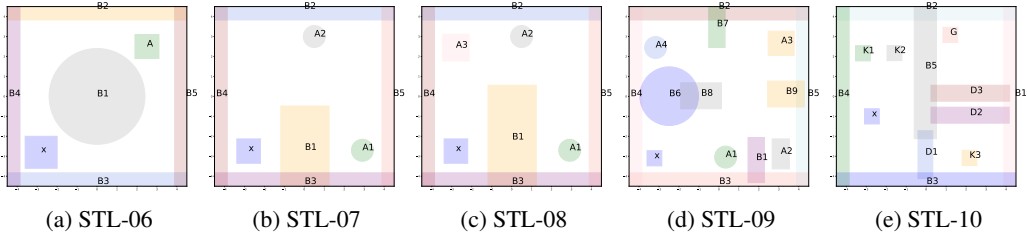

Figure 15: Scene for Unicycle: STL tasks 06 to 10

**STL-06 (Multi-layer):** $F_{[10:90]}(A) \wedge G_{[0:100]}(\neg B_1) \wedge G_{[0:100]}(\neg B_2) \wedge G_{[0:100]}(\neg B_3) \wedge G_{[0:100]}(\neg B_4) \wedge G_{[0:100]}(\neg B_5)$

**STL-07 (Multi-layer):** $F_{[0:90]}(A_1) \wedge F_{[40:80]}(A_2) \wedge G_{[0:100]}(\neg B_1) \wedge G_{[0:100]}(\neg B_2) \wedge G_{[0:100]}(\neg B_3) \wedge G_{[0:100]}(\neg B_4) \wedge G_{[0:100]}(\neg B_5)$

**STL-08 (Multi-layer):** $F_{[0:90]}(A_1) \wedge F_{[40:80]}(A_2 \wedge F_{[10:20]}(G_{[0:10]}(A_3))) \wedge G_{[0:100]}(\neg B_1) \wedge G_{[0:100]}(\neg B_2) \wedge G_{[0:100]}(\neg B_3) \wedge G_{[0:100]}(\neg B_4) \wedge G_{[0:100]}(\neg B_5)$

**STL-09 (Multi-layer):** $F_{[5:20]}(A_1 \wedge F_{[10:20]}(G_{[0:5]}(A_2) \wedge F_{[10:30]}(G_{[0:5]}(A_3)) \wedge F_{[10:30]}(G_{[0:10]}(A_4)))) \wedge G_{[0:100]}(\neg B_1) \wedge G_{[0:100]}(\neg B_2) \wedge G_{[0:100]}(\neg B_3) \wedge G_{[0:100]}(\neg B_4) \wedge G_{[0:100]}(\neg B_5) \wedge G_{[0:100]}(\neg B_6) \wedge G_{[0:100]}(\neg B_7) \wedge G_{[0:100]}(\neg B_8) \wedge G_{[0:100]}(\neg B_9)$

**STL-10 (Multi-layer):** $(\neg D_1)U_{[0:100]}(K_1) \wedge (\neg D_2)U_{[0:100]}(K_2) \wedge (\neg D_3)U_{[0:100]}(K_3) \wedge F_{[80:90]}(G_{[0:5]}(G)) \wedge G_{[0:100]}(\neg B_1) \wedge G_{[0:100]}(\neg B_2) \wedge G_{[0:100]}(\neg B_3) \wedge G_{[0:100]}(\neg B_4) \wedge G_{[0:100]}(\neg B_5)$

### A.10.3 STLs in "Franka Panda" environment

**STL-01 (Two-layer):** $F_{[5:7]}(F_{[50:85]}(A) \wedge G_{[0:90]}(\neg B_5) \wedge G_{[0:100]}(\neg W_1) \wedge G_{[0:100]}(\neg W_2) \wedge G_{[0:100]}(\neg W_3) \wedge G_{[0:100]}(\neg W_4) \wedge G_{[0:100]}(\neg W_5) \wedge G_{[0:100]}(\neg W_6))$

**STL-02 (Two-layer):** $F_{[5:10]}(F_{[0:50]}(A) \wedge G_{[60:80]}(C) \wedge G_{[0:90]}(\neg B_5) \wedge G_{[0:100]}(\neg W_1) \wedge G_{[0:100]}(\neg W_2) \wedge G_{[0:100]}(\neg W_3) \wedge G_{[0:100]}(\neg W_4) \wedge G_{[0:100]}(\neg W_5) \wedge G_{[0:100]}(\neg W_6))$

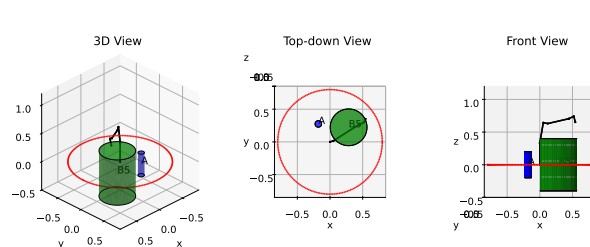

Figure 16: Scene for Franka Panda: STL task 01

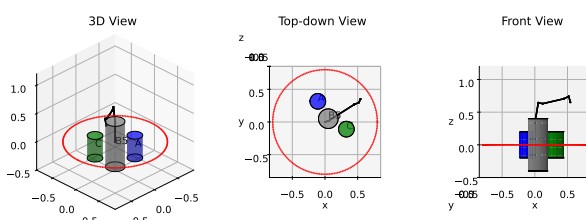

Figure 17: Scene for Franka Panda: STL task 02

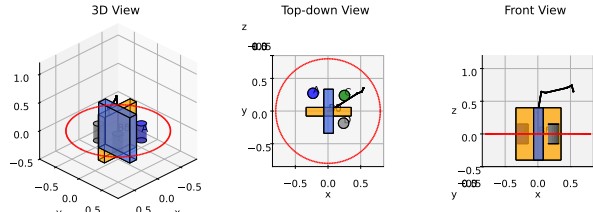

Figure 18: Scene for Franka Panda: STL task 03

**STL-03 (Two-layer):** $F_{[5:10]}(F_{[0:50]}(A) \wedge F_{[40:60]}(C) \wedge G_{[70:80]}(D) \wedge G_{[0:90]}(\neg B_5) \wedge G_{[0:90]}(\neg B_0) \wedge G_{[0:100]}(\neg W_1) \wedge G_{[0:100]}(\neg W_2) \wedge G_{[0:100]}(\neg W_3) \wedge G_{[0:100]}(\neg W_4) \wedge G_{[0:100]}(\neg W_5) \wedge G_{[0:100]}(\neg W_6))$

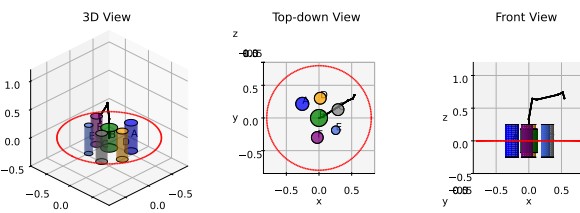

Figure 19: Scene for Franka Panda: STL task 04

**STL-04 (Two-layer):** $F_{[5:10]}(F_{[0:50]}(A) \wedge F_{[40:50]}(C) \wedge F_{[70:80]}(F) \wedge G_{[50:60]}(D) \wedge G_{[0:90]}(\neg B) \wedge G_{[0:90]}(\neg E) \wedge G_{[0:100]}(\neg W_1) \wedge G_{[0:100]}(\neg W_2) \wedge G_{[0:100]}(\neg W_3) \wedge G_{[0:100]}(\neg W_4) \wedge G_{[0:100]}(\neg W_5) \wedge G_{[0:100]}(\neg W_6))$

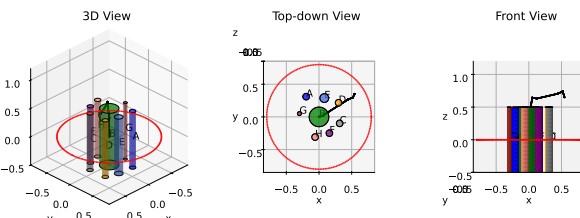

Figure 20: Scene for Franka Panda: STL task 05

**STL-05 (Two-layer):** $F_{[5:10]}(F_{[0:30]}(A) \wedge F_{[30:50]}(C) \wedge F_{[70:80]}(F) \wedge F_{[75:88]}(H) \wedge G_{[50:60]}(D) \wedge G_{[0:90]}(\neg B) \wedge G_{[0:90]}(\neg E) \wedge G_{[0:90]}(\neg G) \wedge G_{[0:100]}(\neg W_1) \wedge G_{[0:100]}(\neg W_2) \wedge G_{[0:100]}(\neg W_3) \wedge G_{[0:100]}(\neg W_4) \wedge G_{[0:100]}(\neg W_5) \wedge G_{[0:100]}(\neg W_6))$

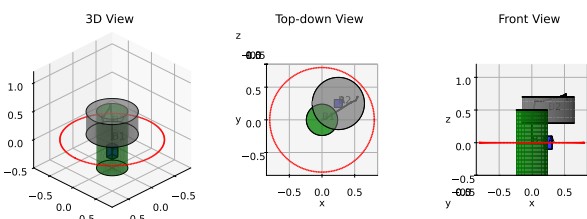

Figure 21: Scene for Franka Panda: STL task 06

**STL-06 (Multi-layer):** $F_{[10:90]}(A) \wedge G_{[0:100]}(\neg B_1) \wedge G_{[0:100]}(\neg B_2) \wedge G_{[0:100]}(\neg W_1) \wedge G_{[0:100]}(\neg W_2) \wedge G_{[0:100]}(\neg W_3) \wedge G_{[0:100]}(\neg W_4) \wedge G_{[0:100]}(\neg W_5) \wedge G_{[0:100]}(\neg W_6)$

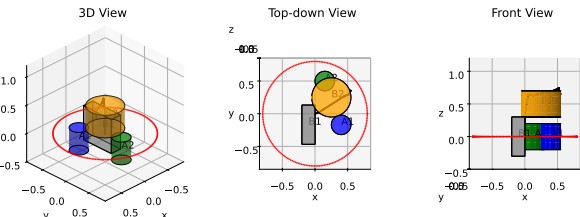

Figure 22: Scene for Franka Panda: STL task 07

**STL-07 (Multi-layer):** $F_{[0:90]}(A_1) \wedge F_{[40:80]}(A_2) \wedge G_{[0:100]}(\neg B_1) \wedge G_{[0:100]}(\neg B_2) \wedge$ $G_{[0:100]}(\neg W_1) \wedge G_{[0:100]}(\neg W_2) \wedge G_{[0:100]}(\neg W_3) \wedge G_{[0:100]}(\neg W_4) \wedge G_{[0:100]}(\neg W_5) \wedge G_{[0:100]}(\neg W_6)$

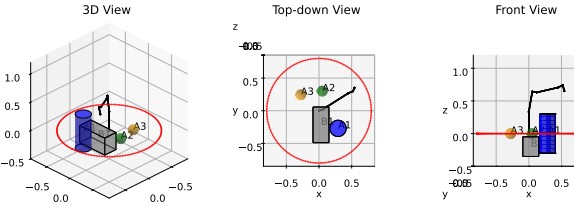

Figure 23: Scene for Franka Panda: STL task 08

**STL-08 (Multi-layer):** $F_{[0:90]}(A_1) \wedge F_{[40:60]}(A_2 \wedge F_{[15:30]}(G_{[0:5]}(A_3))) \wedge G_{[0:100]}(\neg B_1) \wedge$ $G_{[0:100]}(\neg W_1) \wedge G_{[0:100]}(\neg W_2) \wedge G_{[0:100]}(\neg W_3) \wedge G_{[0:100]}(\neg W_4) \wedge G_{[0:100]}(\neg W_5) \wedge G_{[0:100]}(\neg W_6)$

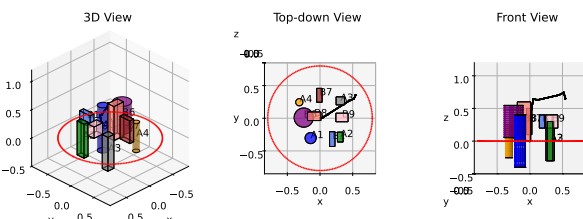

Figure 24: Scene for Franka Panda: STL task 09

**STL-09 (Multi-layer):** $F_{[25:30]}(A_1 \wedge F_{[20:28]}(G_{[0:5]}(A_2) \wedge F_{[10:30]}(G_{[0:5]}(A_3)) \wedge$ $F_{[10:30]}(G_{[0:10]}(A_4)))) \wedge G_{[0:100]}(\neg B_1) \wedge G_{[0:100]}(\neg B_6) \wedge G_{[0:100]}(\neg B_7) \wedge G_{[0:100]}(\neg B_8) \wedge$ $G_{[0:100]}(\neg B_9) \wedge G_{[0:100]}(\neg W_1) \wedge G_{[0:100]}(\neg W_2) \wedge G_{[0:100]}(\neg W_3) \wedge G_{[0:100]}(\neg W_4) \wedge G_{[0:100]}(\neg W_5) \wedge$ $G_{[0:100]}(\neg W_6)$

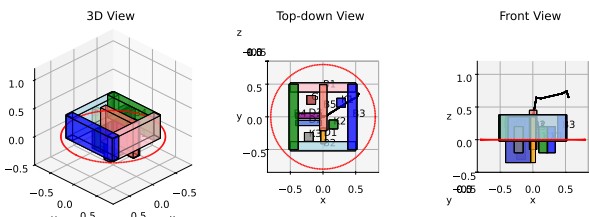

Figure 25: Scene for Franka Panda: STL task 10

**STL-10 (Multi-layer):** $(\neg D_1) U_{[0:100]}(K_1) \wedge (\neg D_2) U_{[0:100]}(K_2) \wedge (\neg D_3) U_{[0:100]}(K_3) \wedge$ $F_{[80:90]}(G_{[0:5]}(G)) \wedge G_{[0:100]}(\neg B_5) \wedge G_{[0:100]}(\neg B_1) \wedge G_{[0:100]}(\neg B_2) \wedge G_{[0:100]}(\neg B_3) \wedge$ $G_{[0:100]}(\neg B_4) \wedge G_{[0:100]}(\neg W_1) \wedge G_{[0:100]}(\neg W_2) \wedge G_{[0:100]}(\neg W_3) \wedge G_{[0:100]}(\neg W_4) \wedge G_{[0:100]}(\neg W_5) \wedge$ $G_{[0:100]}(\neg W_6)$

### A.10.4 STLs in "Quadrotor" environment

**STL-01 (Two-layer):** $F_{[5:7]}(F_{[50:85]}(A) \wedge G_{[0:90]}(\neg B_5) \wedge G_{[0:100]}(\neg W_1) \wedge G_{[0:100]}(\neg W_2) \wedge$ $G_{[0:100]}(\neg W_3) \wedge G_{[0:100]}(\neg W_4) \wedge G_{[0:100]}(\neg W_5) \wedge G_{[0:100]}(\neg W_6))$

**STL-02 (Two-layer):** $F_{[5:10]}(F_{[0:50]}(A) \wedge G_{[60:80]}(C) \wedge G_{[0:90]}(\neg B_5) \wedge G_{[0:100]}(\neg W_1) \wedge$ $G_{[0:100]}(\neg W_2) \wedge G_{[0:100]}(\neg W_3) \wedge G_{[0:100]}(\neg W_4) \wedge G_{[0:100]}(\neg W_5) \wedge G_{[0:100]}(\neg W_6))$

**STL-03 (Two-layer):** $F_{[5:10]}(F_{[0:50]}(A) \wedge F_{[40:60]}(C) \wedge G_{[70:80]}(D) \wedge G_{[0:90]}(\neg B_5) \wedge$ $G_{[0:90]}(\neg B_0) \wedge G_{[0:100]}(\neg W_1) \wedge G_{[0:100]}(\neg W_2) \wedge G_{[0:100]}(\neg W_3) \wedge G_{[0:100]}(\neg W_4) \wedge G_{[0:100]}(\neg W_5) \wedge$ $G_{[0:100]}(\neg W_6))$

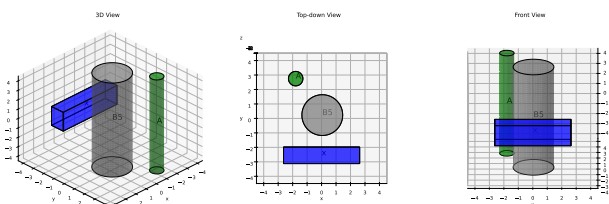

Figure 26: Scene for Quadrotor: STL task 01

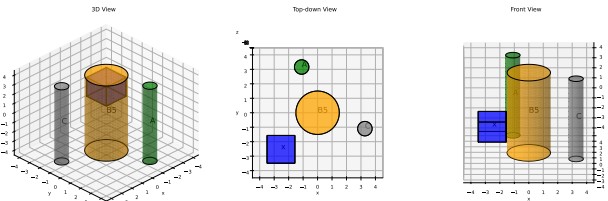

Figure 27: Scene for Quadrotor: STL task 02

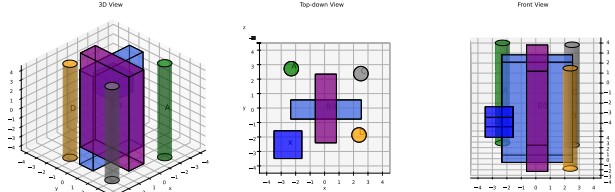

Figure 28: Scene for Quadrotor: STL task 03

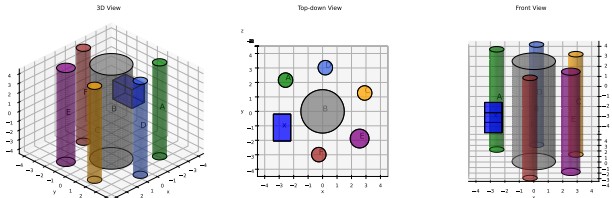

Figure 29: Scene for Quadrotor: STL task 04

**STL-04 (Two-layer):** $F_{[5:10]}(F_{[0:50]}(A) \wedge F_{[40:50]}(C) \wedge F_{[70:80]}(F) \wedge G_{[50:60]}(D) \wedge G_{[0:90]}(\neg B) \wedge G_{[0:90]}(\neg E) \wedge G_{[0:100]}(\neg W_1) \wedge G_{[0:100]}(\neg W_2) \wedge G_{[0:100]}(\neg W_3) \wedge G_{[0:100]}(\neg W_4) \wedge G_{[0:100]}(\neg W_5) \wedge G_{[0:100]}(\neg W_6))$

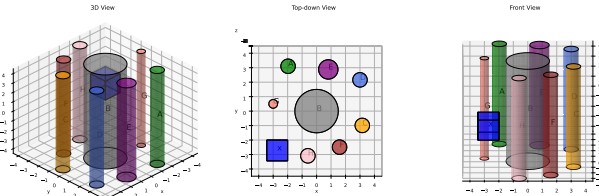

Figure 30: Scene for Quadrotor: STL task 05

**STL-05 (Two-layer):** $F_{[5:10]}(F_{[0:30]}(A) \wedge F_{[30:50]}(C) \wedge F_{[70:80]}(F) \wedge F_{[75:88]}(H) \wedge G_{[50:60]}(D) \wedge G_{[0:90]}(\neg B) \wedge G_{[0:90]}(\neg E) \wedge G_{[0:90]}(\neg G) \wedge G_{[0:100]}(\neg W_1) \wedge G_{[0:100]}(\neg W_2) \wedge G_{[0:100]}(\neg W_3) \wedge G_{[0:100]}(\neg W_4) \wedge G_{[0:100]}(\neg W_5) \wedge G_{[0:100]}(\neg W_6))$

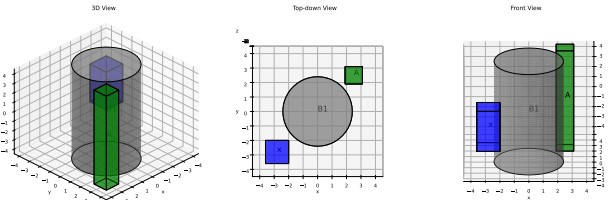

Figure 31: Scene for Quadrotor: STL task 06

**STL-06 (Multi-layer):** $F_{[10:90]}(A) \wedge G_{[0:100]}(\neg B_1) \wedge G_{[0:100]}(\neg W_1) \wedge G_{[0:100]}(\neg W_2) \wedge G_{[0:100]}(\neg W_3) \wedge G_{[0:100]}(\neg W_4) \wedge G_{[0:100]}(\neg W_5) \wedge G_{[0:100]}(\neg W_6)$

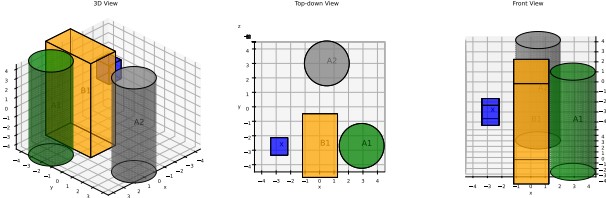

Figure 32: Scene for Quadrotor: STL task 07

**STL-07 (Multi-layer):** $F_{[0:90]}(A_1) \wedge F_{[40:80]}(A_2) \wedge G_{[0:100]}(\neg B_1) \wedge G_{[0:100]}(\neg W_1) \wedge G_{[0:100]}(\neg W_2) \wedge G_{[0:100]}(\neg W_3) \wedge G_{[0:100]}(\neg W_4) \wedge G_{[0:100]}(\neg W_5) \wedge G_{[0:100]}(\neg W_6)$

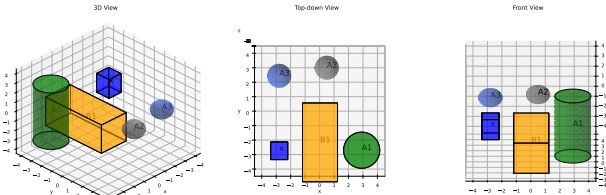

Figure 33: Scene for Quadrotor: STL task 08

**STL-08 (Multi-layer):** $F_{[0:90]}(A_1) \wedge F_{[40:60]}(A_2 \wedge F_{[15:30]}(G_{[0:5]}(A_3))) \wedge G_{[0:100]}(\neg B_1) \wedge G_{[0:100]}(\neg W_1) \wedge G_{[0:100]}(\neg W_2) \wedge G_{[0:100]}(\neg W_3) \wedge G_{[0:100]}(\neg W_4) \wedge G_{[0:100]}(\neg W_5) \wedge G_{[0:100]}(\neg W_6)$

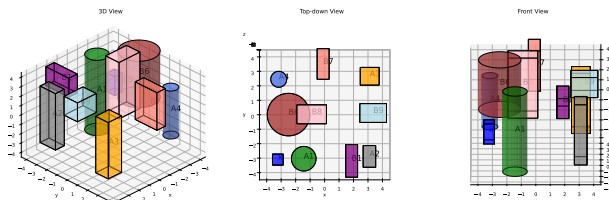

Figure 34: Scene for Quadrotor: STL task 09

**STL-09 (Multi-layer):** $F_{[25:30]}(A_1 \land F_{[20:28]}(G_{[0:5]}(A_2) \land F_{[10:30]}(G_{[0:5]}(A_3)) \land F_{[10:30]}(G_{[0:10]}(A_4)))) \land G_{[0:100]}(\neg B_1) \land G_{[0:100]}(\neg B_6) \land G_{[0:100]}(\neg B_7) \land G_{[0:100]}(\neg B_8) \land G_{[0:100]}(\neg B_9) \land G_{[0:100]}(\neg W_1) \land G_{[0:100]}(\neg W_2) \land G_{[0:100]}(\neg W_3) \land G_{[0:100]}(\neg W_4) \land G_{[0:100]}(\neg W_5) \land G_{[0:100]}(\neg W_6)$

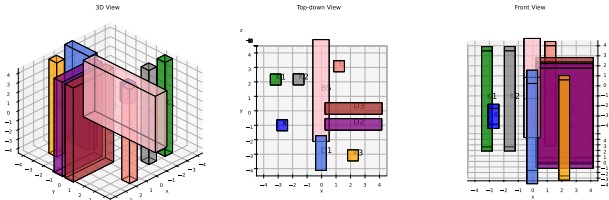

Figure 35: Scene for Quadrotor: STL task 10

**STL-10 (Multi-layer):** $(\neg D_1)U_{[0:100]}(K_1) \land (\neg D_2)U_{[0:100]}(K_2) \land (\neg D_3)U_{[0:100]}(K_3) \land F_{[80:90]}(G_{[0:5]}(G)) \land G_{[0:100]}(\neg B_5) \land G_{[0:100]}(\neg W_1) \land G_{[0:100]}(\neg W_2) \land G_{[0:100]}(\neg W_3) \land G_{[0:100]}(\neg W_4) \land G_{[0:100]}(\neg W_5) \land G_{[0:100]}(\neg W_6)$

### A.10.5 STLs in "Ant" environment

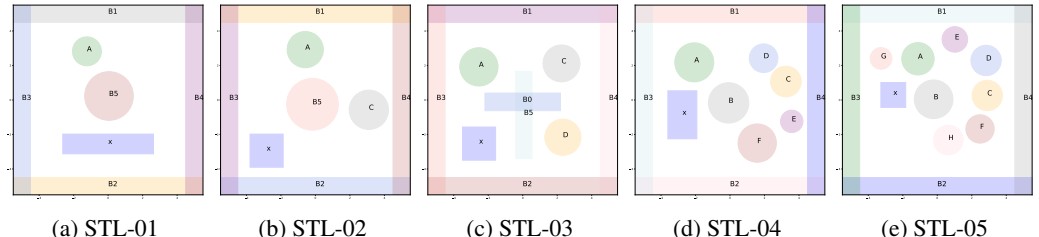

(a) STL-01     (b) STL-02     (c) STL-03     (d) STL-04     (e) STL-05

Figure 36: Scene for Ant: STL tasks 01 to 05

**STL-01 (Two-layer):** $F_{[10:14]}(F_{[100:170]}(A) \land G_{[0:180]}(\neg B_5) \land G_{[0:180]}(\neg B_1) \land G_{[0:180]}(\neg B_2) \land G_{[0:180]}(\neg B_3) \land G_{[0:180]}(\neg B_4))$

**STL-02 (Two-layer):** $F_{[10:20]}(F_{[0:100]}(A) \land G_{[120:160]}(C) \land G_{[0:180]}(\neg B_5) \land G_{[0:180]}(\neg B_1) \land G_{[0:180]}(\neg B_2) \land G_{[0:180]}(\neg B_3) \land G_{[0:180]}(\neg B_4))$

**STL-03 (Two-layer):** $F_{[10:20]}(F_{[0:100]}(A) \land F_{[80:120]}(C) \land G_{[140:160]}(D) \land G_{[0:180]}(\neg B_5) \land G_{[0:180]}(\neg B_0) \land G_{[0:180]}(\neg B_1) \land G_{[0:180]}(\neg B_2) \land G_{[0:180]}(\neg B_3) \land G_{[0:180]}(\neg B_4))$

**STL-04 (Two-layer):** $F_{[10:20]}(F_{[0:100]}(A) \land F_{[80:100]}(C) \land F_{[140:160]}(F) \land G_{[100:120]}(D) \land G_{[0:180]}(\neg B) \land G_{[0:180]}(\neg E) \land G_{[0:180]}(\neg B_1) \land G_{[0:180]}(\neg B_2) \land G_{[0:180]}(\neg B_3) \land G_{[0:180]}(\neg B_4))$

**STL-05 (Two-layer):** $F_{[10:20]}(F_{[0:60]}(A) \land F_{[60:100]}(C) \land F_{[140:160]}(F) \land F_{[150:176]}(H) \land G_{[100:120]}(D) \land G_{[0:180]}(\neg B) \land G_{[0:180]}(\neg E) \land G_{[0:180]}(\neg G) \land G_{[0:180]}(\neg B_1) \land G_{[0:180]}(\neg B_2) \land G_{[0:180]}(\neg B_3) \land G_{[0:180]}(\neg B_4))$

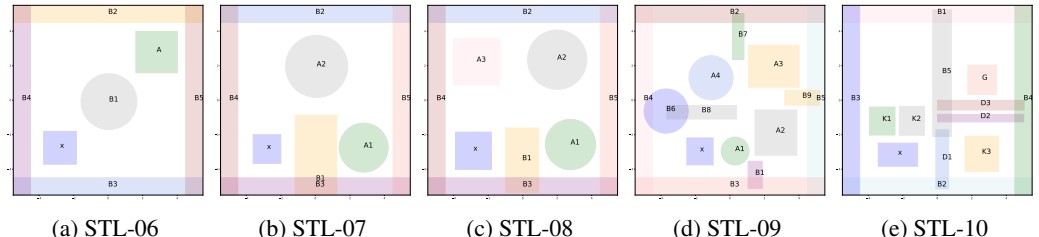

(a) STL-06     (b) STL-07     (c) STL-08     (d) STL-09     (e) STL-10

Figure 37: Scene for Ant: STL tasks 06 to 10

**STL-06 (Multi-layer):** $F_{[20:180]}(A) \wedge G_{[0:200]}(\neg B_1) \wedge G_{[0:200]}(\neg B_2) \wedge G_{[0:200]}(\neg B_3) \wedge G_{[0:200]}(\neg B_4) \wedge G_{[0:200]}(\neg B_5)$

**STL-07 (Multi-layer):** $F_{[0:180]}(A_1) \wedge F_{[80:160]}(A_2) \wedge G_{[0:200]}(\neg B_1) \wedge G_{[0:200]}(\neg B_2) \wedge G_{[0:200]}(\neg B_3) \wedge G_{[0:200]}(\neg B_4) \wedge G_{[0:200]}(\neg B_5)$

**STL-08 (Multi-layer):** $F_{[0:180]}(A_1) \wedge F_{[80:160]}(A_2 \wedge F_{[20:40]}(G_{[0:20]}(A_3))) \wedge G_{[0:200]}(\neg B_1) \wedge G_{[0:200]}(\neg B_2) \wedge G_{[0:200]}(\neg B_3) \wedge G_{[0:200]}(\neg B_4) \wedge G_{[0:200]}(\neg B_5)$

**STL-09 (Multi-layer):** $F_{[10:40]}(A_1 \wedge F_{[20:40]}(G_{[0:10]}(A_2) \wedge F_{[20:60]}(G_{[0:10]}(A_3)) \wedge F_{[20:60]}(G_{[0:20]}(A_4)))) \wedge G_{[0:200]}(\neg B_1) \wedge G_{[0:200]}(\neg B_2) \wedge G_{[0:200]}(\neg B_3) \wedge G_{[0:200]}(\neg B_4) \wedge G_{[0:200]}(\neg B_5) \wedge G_{[0:200]}(\neg B_6) \wedge G_{[0:200]}(\neg B_7) \wedge G_{[0:200]}(\neg B_8) \wedge G_{[0:200]}(\neg B_9)$

**STL-10 (Multi-layer):** $(\neg D_1)U_{[0:200]}(K_1) \wedge (\neg D_2)U_{[0:200]}(K_2) \wedge (\neg D_3)U_{[0:200]}(K_3) \wedge F_{[160:180]}(G_{[0:10]}(G)) \wedge G_{[0:200]}(\neg B_1) \wedge G_{[0:200]}(\neg B_2) \wedge G_{[0:200]}(\neg B_3) \wedge G_{[0:200]}(\neg B_4) \wedge G_{[0:200]}(\neg B_5)$

## A.11 POSSIBLE SOLUTIOSN REGARDING THE LIMITATIONS

Here we discuss two directions to address the limitations mentioned in Sec. 5.6 for future work:

**LLM-based logic weighting / pruning.** Right now, all the time variable assignments are treated equally on the same level. To better handle (multiple) disjunction(s), we propose leveraging Large Language Models (LLMs) as a high-level logical prior. An LLM can reason over the semantic structure of the STL formula to weigh or prune unlikely branches of the logical graph and select the most promising disjunctive branch. This would reduce the problem from sampling over all logical possibilities to a single (or a subset of) high-likelihood branches.

**Time variable-event binding for solution sparsity.** A key limitation in scaling to the "Always Eventually" (GF) operators is the redundancy of time variables. For an STL like $G_{[0,200]}F_{[10,30]}$Reach(A) a naive STL decomposition algorithm assigns a distinct time variable for every time step $t \in [0, 200]$, but in fact these time variables are highly correlated. A single "Reach(A)" event at $t = 35$ can be used for all time variables $t \in [5, 25]$ for the outer "G" operator. Future work might consider working on a sparse set of K event anchors (or a partition for the time axis), where each anchor (partition) can bind with multiple time variables for the outer operator, which will effectively narrow the assignment complexity towards the number of sparse events. Constructing the event space and establishing the association between the event and time variables is left as an open question, with potential solutions including a learnable attention mechanism, hand-crafted and heuristic-based temporal clustering, or using an LLM to provide suggestions.

