# OpenReview forum: "TGPO: Temporal Grounded Policy Optimization for Signal Temporal Logic Tasks"
_ICLR.cc/2026/Conference — Submitted to ICLR 2026_

### Official Review · Reviewer_TtqM · 2025-10-30

**Soundness:** 2
**Presentation:** 2
**Contribution:** 3
**Rating:** 2
**Confidence:** 3

**Summary:**

This paper presents a reinforcement learning (RL) approach for learning from signal temporal logic (STL) to make learning more feasible for long-horizon tasks. The novel model-free approach divides and flattens complex STL formulas and searches for time-variable actualizations via Metropolis-Hastings (MH) sampling to enable efficient learning. The proposed method is compared with a range of existing approaches across several environments. I believe the idea is original and shows promise for improving over existing methods for STL learning. However, the paper still needs substantial work; specifically, a more thorough technical analysis and a systematic description of the proposed approach, as well as clearer explanations and presentation of the experimental results.

**Strengths:**

To the best of my knowledge, the proposed approach is a novel way to address the sparsity and non-Markovian nature of STL rewards, an important problem in STL learning that existing methods do not sufficiently handle, especially for long-horizon tasks with complex STL formulas. In addition, the proposed approach is compared against a sufficiently representative set of baselines.

**Weaknesses:**

- **W1.** (Minor) Eq. (2) can be visually improved.
- **W2.** The problem statement is quite vague and somewhat confusing. I think a more comprehensive and consistent notation can be used. Also, constructing an augmented MDP should be part of the solution rather than the objective. Lastly, the probability-maximization statement with respect to the initial state is also ambiguous.
- **W3.** The STL decomposition is not systematically or algorithmically explained (Algorithm 1 is very high level) and is presented primarily through intuition and examples.
- **W4.** The number of different time-variable assignments can grow exponentially with the number of subgoals, which can be further complicated for nested and/or disjunctive formulas.
- **W5.** I think observed (continuous) time should be part of the augmented state space rather than the discrete step number, as STL is defined for continuous signals.
- **W6.** Exact time-variable assignments do not seem realistic; as I understand it, they represent the exact time at which a certain state must be reached.
- **W7.** It seems that a new MDP is constructed for every new assignment, which can technically impede RL.
- **W8.** The formal connection between the dense rewards and STL satisfaction is not established or discussed.
- **W9.** The authors do not provide sufficient technical analysis of the time-variable assignment procedure.
- **W10.** The flow and clarity of the experiments section can be improved.

**Questions:**

- **Q1.** See W2. What does $\mathbb{P}_{x_0\in\mathcal{X}_0}$ formally mean in the problem statement? Do we assume a distribution over the initial states? Is the objective maximization over success rates or average robustness score?
- **Q2.** See W3. Can you provide pseudocode for STL decomposition? What are the inputs and outputs? How are nested formulas formally divided? How are the time variables determined; for example, for $G_{[0,200]} F_{[0,10]} a$, where the number of time variables can vary, or for $F_{[0,100]} a \vee F_{[0,100]} b$, where one time variable might be sufficient?
- **Q3.** What are your thoughts regarding W4?
- **Q4.** See W6. Why isn't an ordering of subgoals sufficient?
- **Q5.** See W7. What are your thoughts on such a nonstationary environment? Why do you think your procedure transfers knowledge efficiently across MDPs?
- **Q6.** See W8. Does maximizing the return maximize the STL success rate or the robustness score? Is it an admissible heuristic?
- **Q7.** See W9. For example, assuming deep RL converges to an optimal policy that maximizes return, is the time-variable assignment procedure complete; i.e., if there is a feasible assignment, will the procedure eventually propose such an assignment? How fast could that be, given W4?
- **Q8.** See W9. Could you provide pseudocode for time-variable assignment with MH sampling? Also, why do you think the critic provides a good heuristic for MH sampling given that the MDPs are changing?
- **Q9.** I think some of the STL formulas used in the experiments should be discussed in the main paper. Could you provide a general description of the STL formulas?
- **Q10.** In the figures in the experiments section, are training steps the same as environment steps?
- **Q11.** Could you compare the dimensionality of your augmented state space with history-stacked state spaces?
- **Q12.** Regarding Figure 6, did you use the same STL formulas with the same time intervals? Could you provide similar figures for different environments where the intervals are also scaled accordingly?

---

> ### Author Response · Authors · 2025-11-21
>
> We thank the reviewer for recognizing our novel approach to handling the sparsity and non-Markovian nature of STL rewards, especially for long-horizon tasks with complex formulas. We also appreciate the positive assessment of our comprehensive experiments compared against a sufficiently representative set of baselines. Below, we provide a detailed discussion of the weaknesses (W) and questions (Q) raised.
>
> > W1:  (Minor) Eq. (2) can be visually improved.
>
> Thanks for the suggestion on the writing. We have updated the manuscript with Eq. (2) visually improved.
>
> > W2/Q1: Clarification on the problem statement.
>
> We have modified the manuscript to have a clearer problem statement. (Section 3.3) We assume there is a distribution for the initial states on a set. The objective is to maximize over success rates for these initial states.
>
> > W3/Q2: STL decomposition is not systematically or algorithmically explained.
>
> We have updated the paper to introduce the STL decomposition algorithm (Section 4.1, Algorithm 1). For $G_{[0,200]}F_{[0,10]} a$, we have 1 time variable for each index in the time window for G. And for $F_{[0,100]}a \lor F_{[0,100]} b$, although we haven't tackled them in this paper, theoretically we can use a binary variable to indicate the branch of the conditions, and then introduce a time variable under each branch, and use the binary variable to "activate" the corresponding time variable.
>
> > W4/Q3: The number of different time-variable assignments can grow exponentially with the number of subgoals, which can be further complicated for nested and/or disjunctive formulas.
>
> We thank the reviewer for raising the important issue of combinatorial complexity. While we briefly mention this in our "limitation" section in the original manuscript, we argue that the practical problem in the STL domain remains tractable for our TGPO for two reasons.
>
> 1. **Density of feasible solutions.** In robotics tasks, feasible time assignments typically form continuous, dense volumes rather than isolated points (which is shown in our Figure 7). Due to the continuity of system dynamics, if a task is solvable at time $t$, it is usually solvable within a range $[t-\Delta_1, t+\Delta_2]$. This implies that the set of valid assignments constitutes a significant portion of the search space. Consequently, even uniform sampling has a high probability of identifying a feasible allocation, and our Bayesian sampling can further improve the efficiency by exploiting the gradients of the learned value landscape to rapidly focus the search on these feasible regions.
>
> 2. **Advancement over existing RL baselines**. Critically, the complexity of these temporal tasks must be viewed relative to the capabilities of existing baselines. For baselines like F-MDP or $\tau$-MDP struggle to solve STL with more than two temporal layers or long-horizon tasks (T>300). In contrast, TGPO can solve multi-level nested tasks on high-dimensional systems. This demonstrates that our hierarchical grounding approach significantly expands the frontier of STL specifications that are tractable for RL methods.
>
> We acknowledge that, similar to other sampling-based motion planning algorithms (e.g., RRT), our efficiency scales inversely with the dimension of the sampling space. However, this is a trade-off that is hard to avoid when solving problems involving complex dynamics where MILP or gradient-based solvers are inapplicable. To address this in future work, we plan to investigate the integration of commonsense priors from LLMs to prune the search space by ruling out semantically infeasible assignments.
>
>
>
> > W5:  The observed (continuous) time should be in the augmented state rather than the discrete step number, as STL is defined for continuous signals.
>
> Since we consider discrete-time dynamical systems, including the discrete step number in the state provides the neural network with identical information as the continuous time; the distinction is merely a scalar constant (time duration). Furthermore, in implementation, we normalized the features to ensure that numerically the input is in the reasonable range. We have made it clear in the updated paper (Section 4.2, between Eq(4) and Eq(5)).

---

> ### Author Response · Authors · 2025-11-21
>
> > W6/Q4: Exact time-variable assignments do not seem realistic, as they require reaching at the exact time./ Why isn't an ordering of subgoals sufficient?
>
> Our time-variable assignment, state transition and reward design does not require the agent to reach a subgoal "only" at an exact time $\tau$, instead, it requires the agent to reach the subgoal "at least" at $\tau$ (the agent can reach the subgoal before $\tau$ but it needs to also be there at $\tau$.) which makes the requirement not that hard to meet. Just specifying the order of the subgoals is not sufficient, as it does not keep track of the subgoal achieving status, and then defining transition and the reward can be a challenge. Meanwhile, since now there is no explicit condition in the augmented state space regarding the "pace", the environment becomes non-stationary and makes the training less stable (for STL $F_{[5,100]} A \land F_{[80,90]} B$, for the order A->B, one possible $t_A$ should be in [5, 79] and $t_B$ in [80,90], then during training the policy might change from faster-ones ($t_A$=5) to slower-ones ($t_A$=75), but since we don't have this $t$ in context, the neural network cannot distinguish between these two, and this will cause non-stationary RL and make training less stable.) Thus we use the exact time-variable conditions to have more controllable behavior and more stable training.
>
>
> > W7/Q5: It seems that a new MDP is constructed for every new assignment, which can technically impede RL.
>
> Our TGPO falls into the category of a stationary contextual MDP[1], where we include time variable assignment $\mathbf{t}$ as the context. In this augmented state space, the reward function and the transition dynamics are fixed functions of the state and $\mathbf{t}$ (won't drift over time). The RL agent is not "re-learning" a new MDP each time. Instead, it is learning a mapping from contexts ($\mathbf{t}$) to behaviors. This stationary setup and the fact that similar time variable assignments often lead to similar policies (unless changing subgoal orders) allow efficient knowledge transfer among assignments.
>
>
>
> > W8/Q6: Lack formal connection between the dense rewards and STL satisfaction (does maximizing the return maximize the STL success rate or the robustness score?) Is it an admissible heuristic?
>
> We have updated the manuscript with an formal analysis (in appendix A.5) linking our designed dense reward to STL satisfaction. We show that, with certain scaling relationship among the reward terms, for a given initial state, if the problem is feasible, the optimal policy can satisfy the STL task. Thus, maximizing the expected return here is equivalent to maximizing the STL success rate over the distribution of initial states.
>
> Regarding the "admissible heuristic", we respectfully clarify that this terminology (derived from informed search algorithm like A\*) does not directly apply to our RL framework, as the training process does not involve heuristic search over the state space. However, if the reviewer's concern is whether the dense rewards might bias the agent away from the STL satisfaction, as we show in the updated manuscript (Appendix A.5), the optimal policy always maximize the STL success rate.
>
> **References**
> 1. Hallak, Assaf, Dotan Di Castro, and Shie Mannor. "Contextual markov decision processes." arXiv preprint arXiv:1502.02259 (2015).

---

> > ### Author Response · Authors · 2025-11-21
> >
> > > W9/Q7/Q8: technical analysis (completeness; why critic is a good heuristic) and pseudocode for the time-variable assignment sampling procedure.
> >
> > **Completeness analysis**: Our algorithm provides a guarantee for probabilistic completeness. Since the space of discrete time-variable assigments is finite, and our hybrid sampler includes a uniform sampling component (40%), every valid assigment has a non-zero probability of being sampled. In the limit of infinite samples, every feasible assignment is guaranteed to be visited infinitely often, providing the RL agent sufficient data and iterations to converge to the optimal policy for that specific assigment.
> >
> > **Critic as heuristic**: The critic estimates the expected return of the policy given a specific time-variable assignment. In our paper, the return includes dense stage-wise progression reward. A key insight in our framework is that an assignment allowing more progress (without breaking invariant properties) is often most likely to be feasible. A physically impossible assignment forces the agent to fail early, resulting in low accumulated reward. Conversely, a feasible (or near-feasible) assignment allows tha agent to complete more subgoals, getting higher returns even before the optimal policy is mastered. Consequently, the critic assigns higher values to these "high-progress" time-variable assignments, effectively serving as a powerful heuristic to guide the MCMC sampler towards the feasible regions where the task can be solved.
> >
> >
> > **Pseudocode**: We respectfully point the reviewer to Algorithm 2 in Appendix A.5 of our original submission (Algorithm 3 in Appendix A.7 in the updated manuscript), which provides the detailed algorithm implementation for time variable MH sampling. We have added a reference to this algorithm in the main text (Sec 4.3, the last sentence) to ensure it is not missed.
> >
> >
> > > Q10: In Figure 5, are the training steps the same as environment steps?
> >
> > We clarify that the "training steps" in Figure 5 refer to training epochs, not environment steps. In each epoch, the policy collects the data by interacting with the simulation environment, and then the policy is updated on the collected data. One training step collects $NT$ environment steps of data, where $N$ is the number of parallel environments ($N=512$ in this paper), and $T$ is the task horizon ($T=100$ in most cases).
> >
> >
> > > Q11: Dimensionality: TGPO vs. history-stacked state space.
> >
> >
> > The augmented state dimension for TGPO is $n+4+N_c$, where $n$ is the original system state dimension, $4$ additional variables to record time index, progress index, previous progress index and the certificate to proceed to the next subgoal, and $N_c$ variables to record satisfaction status for the invariant properties. The dimension for the history-stacked state space is $nT$, where $T$ is the task horizon. For our 12D quadrotor task with a horizon of $T=100$ and 4 invariance constraints, TGPO requires an input dimension of only 20, whereas the history-stacked method will require 1200 dimension input. The massive difference in dimensionality is the main reason why TGPO strikes much better than $\tau$-MDP for long-horizon tasks.
> >
> > > Q12: Clarification for the varied-horizon experiments and additional results on different environments.
> >
> > In our original varied-horizon experiment, the time interval for each subformula is scaled proportionally to the task horizon. Due to time constraint, we conducted the experiments on the unicycle experiment with 3 random seeds and show the result in the appendix A.6. The conclusion is similar to what we had on the linear case, which shows that our approach’s effectiveness in handling long-horizon tasks is consistent across different system dynamics.
> >
> >
> >
> > **We hope our response has addressed your concerns. We kindly ask if there are any remaining issues preventing a more positive evaluation, and we would like to continue the discussion in the remaining rebuttal period.**

---

> > > ### Comment · Reviewer_TtqM · 2025-11-26
> > >
> > > I thank the authors for going over my comments and for their response. I have included my additional comments below:
> > >
> > > **W2/Q1.**
> > > I believe that explicitly mentioning this initial state distribution in Sections 3.2 or 3.3 could improve the quality and clarity of the paper.
> > >
> > > **W3/Q2.**
> > > This means that $G_{[0,200]}F_{[0,10]}$ requires 200 time variables, which is much higher than the 5 variables used in the experiments; it is also still not clear to me how disjunctions are handled. I realized that the experiments did not include disjunctions and GF subformulas, and in the updated manuscript under (2) it is mentioned that such formulas are not considered. This means the proposed approach does not cover the entire STL but only a fragment of it (which also needs to be formally defined), diminishing the contribution of the paper.
> > >
> > > **W4/Q3.**
> > > I do agree that, without disjunctions and GF subformulas, the proposed approach can improve upon existing approaches for a reasonably small number of time variables. However, I still think that exponentially growing time-variable assignments are one of the main limitations of your approach and deserve more than a very brief mention.
> > >
> > > **W8/Q6.**
> > > How do you choose $\lambda_1, \lambda_2, \lambda_3,$ and $\lambda_4$ in Appendix 5?
> > >
> > > **Q9.**
> > > I think it is important to discuss what kinds of STL formulas are included in the experiments. For example, why are STL-06 formulas considered multi-layer?

---

> > > > ### Author Response · Authors · 2025-11-27
> > > >
> > > > We thank the reviewer for the follow-up questions. We have modified the paper accordingly and included our comments below.
> > > >
> > > > **W2/Q1:** Thanks for this feedback. As suggested, we have added this initial state distribution to Section 3.2 to improve the paper's quality.
> > > >
> > > > **W4/Q3 \& W3/Q2:** We thank the reviewer for these comments. We agree that “not handling disjunction, GF, and scalability towards more time variables” are the limitations of TGPO. We have explicitly mentioned this limitation in Sections 5 & 6. In addition, we have removed the word “general” from our contribution, and have **added a new section in Appendix A.11 to discuss potential solutions about how to address these limitations.**
> > > >
> > > > However, we argue that our contribution lies in “proposing a novel STL RL algorithm” (which all the reviewers agree upon) and "solving the unsolved": **TGPO can solve deeply nested STLs in long-horizon tasks under complex system dynamics that previous methods cannot solve or struggle to solve.** In the experiments, we have shown leading performance over 10 STLs on 5 different benchmarks, and in long-horizon cases as well (200-1000 steps). We believe these results constitute a significant step forward for the STL community.
> > > >
> > > > **W8/Q6:** For hyperparameter selection, we ran a hyperparameter search in the “Linear” dynamics environment using a subset of STL tasks and selected the configuration that yielded the highest performance. (To ensure fair comparisons, we applied a similar hyper-parameter tuning process for all the baselines.) From the theoretical analysis shown in Appendix A. 5, a more systematic way to choose the hyperparameter assignment would be: first estimate the lower and upper bounds for the discounted distance-based return $Z$ to compute the $\Delta Z$, and then set $\lambda_1=1$, and find the $\lambda_2, \lambda_3, \lambda_4$ can satisfy the inequality constraints shown in Theorem 1. Since the inequality is homogeneous, these coefficients can be scaled globally to match the magnitude of empirical rewards while strictly preserving the theoretical guarantees.
> > > >
> > > > **Q9:** We have clarified the fragment we can work on (as long as it does not contain GF or disjunction operators) in Section 3.1 (last sentence) and detailed it in Section 5.1 (Benchmark subsection). The types of STLs are mainly for locomotion and manipulation tasks with temporal constraints and with varied system dynamics. We include single-layer STLs like STL-06 in the appendix A 10.1 to complete the scope of STLs TGPO can handle, whereas the F-MDP can only solve two-layer STLs.
> > > >
> > > >
> > > > **We kindly ask if there are any remaining issues preventing a more positive evaluation, and we would like to continue the discussion in the remaining rebuttal period. But anyway, we appreciate the time you have taken to continue reviewing our work during this busy period.**

---

### Official Review · Reviewer_sqA6 · 2025-10-31

**Soundness:** 3
**Presentation:** 3
**Contribution:** 3
**Rating:** 8
**Confidence:** 2

**Summary:**

The paper proposes Temporal Grounded Policy Optimization (TGPO), a hierarchical reinforcement learning framework for solving control problems specified using Signal Temporal Logic (STL). STL enables rich task specifications with temporal and spatial constraints, but its non-Markovian structure and sparse reward signals make it difficult to handle with standard RL algorithms.  TGPO decomposes STL formulas into subgoals with invariant constraints, and introduces a two-level architecture: a high-level “temporal grounding” component assigns time variables to each subgoal, while a low-level time-conditioned policy learns to satisfy them using dense, stage-wise rewards. The framework includes a critic-guided Bayesian time allocation step using Metropolis–Hastings sampling, which focuses exploration on promising temporal schedules.
Experiments across five environments (2D navigation, unicycle, Franka Panda, quadrotor, and Ant) show that TGPO and its Bayesian variant (TGPO*) outperform several baselines—τ-MDP, F-MDP, RNN, Grad, and CEM—particularly on complex, high-dimensional, and long-horizon STL tasks.

**Strengths:**

Strengths
1. Solid statistical analysis and thorough results section.
The empirical evaluation is broad, well-structured, and uses multiple seeds, metrics, and ablation studies. The inclusion of both low- and high-dimensional systems helps demonstrate scalability.
2. The use of the learned value function to guide time allocation search is interesting
3. The writing is clear for the most part (see weakness 1)

**Weaknesses:**

1. In sec. 4.1 the authors say “Our method of decomposing STL into subgoals with invariant constraints is inspired by Kapoor et al. (2024); Liu et al. (2025).”  From a clarity standpoint, it is unclear to what extent the proposed method for decomposition into subgoals differs from these prior works.  Can the authors elaborate on this?
2. The shaped reward defined in eq. 6 seems specific to locomotion-type problems.  It’s unclear how this would generalize to other domains.

**Questions:**

1. See weaknesses 1 and 2
2. In what sense is the time-allocation method Bayesian?
The Metropolis–Hastings sampling uses the critic as a heuristic energy function, but it’s unclear whether this constitutes a Bayesian inference procedure or simply a guided stochastic search.

---

> ### Author Response · Authors · 2025-11-21
>
> We thank the reviewer for the positive feedback, especially regarding our broad empirical evaluation across diverse systems, and the use of the learned value function for time-allocation search. We also appreciate the comments on the clarity of our writing. Below, we provide a detailed discussion of the weaknesses (W) and questions (Q) raised.
>
> > W1, Q1: How TGPO decomposes STL differently compared to prior work.
>
> Decomposing STLs into reachability and invariance tasks is introduced in [1] to accelerate classical solver solving process for long-horizon STL tasks. Then the work in [2] leverages it to conduct zero-shot trajectory planning based on generative models learned from demonstration data.
>
> The novelty of our work is not the decomposition technique itself, but an inherit novel RL algorithm to solve broader class of STL problems without demonstrations: we introduce a dense, stage-wise reward design and an augmented MDP that are directly aligned with each subgoal’s semantics, and we further view the problem through a temporal variable assignment perspective, and use the critic to guide a more efficient search. This leads to improved scalability, interpretability, and the ability to handle more general, nested, and long-horizon STL tasks compared to prior works.
>
> > W2, Q1: Generability to other domains.
>
> We would like to respectfully clarify that TGPO is not limited to locomotion tasks. The Franka Panda benchmark used in our paper is a manipulation task and not a locomotion task. While Eq. 6 uses a distance-based reward, the reward structure is domain-agnostic because it is derived from STL robustness semantics. The high-level reward structure depends on how "far" a trajectory is from satisfying a subgoal, flags for subgoal completion and invariance constraint violation. These components are not tied to locomotion and can be generalized to other domains by defining appropriate task-relatd predicates.
>
> > Q2: In what sense is the time-allocation method Bayesian?
>
> We classify our time-allocation algorithm as Bayesian, because we seek to find the posterior distribution of "good" time variable assignments that can lead to the fulfillment of the STL task. Our prior belief over the time variables follows a uniform distribution over their corresponding intervals. We use the learned critic value as a learned, non-normalized approximation of the log-likelihood of success, and we use MH algorithm to sample time variables from the updated belief at each training epoch. We have clarified this in the revised manuscript (Section 4.3).
>
> **References**
> 1. Kapoor, Parv, Eunsuk Kang, and Rômulo Meira-Góes. "Safe planning through incremental decomposition of signal temporal logic specifications." NASA Formal Methods Symposium. Cham: Springer Nature Switzerland, 2024.
> 2. Liu, Ruijia, et al. "Zero-Shot Trajectory Planning for Signal Temporal Logic Tasks." arXiv preprint arXiv:2501.13457 (2025).

---

### Official Review · Reviewer_G7gi · 2025-11-01

**Soundness:** 2
**Presentation:** 3
**Contribution:** 2
**Rating:** 4
**Confidence:** 4

**Summary:**

This paper presents a new reinforcement learning method to learn control policies for some types of STL specifications. The proposed method consists of first sampling time assignments for decomposed subgoals and then learn policies to achieve these subgoals conditioned on the time assignments.

**Strengths:**

This work presents a new reinforcement learning method for motion planning to achieve sequenced subgoals given time assignments. The formulation of several RL components including state space and reward function seems to be new.

This work proposes a new method for time domain planning in STL that is capable of sampling promising future timesteps for subgoal planning.

Experiments in five simulated environments demonstrate improved performance over some baseline methods.

**Weaknesses:**

Although the paper claims to solve *general* STLs, in fact the proposed method cannot be applied to some important STLs at the moment.
- Line 197, it is assumed that the type of invariance task after translation only contains *interval* for time variable. This makes modelling and learning for STL that involves infinite time impractical. Although this is mentioned in the limitation section, It should be clearly described at the introduction of STLs.
- This paper cannot handle STLs like G(F …) as mentioned in Line 212. In fact, many liveness properties of the STL cannot be handled by the current method. These kinds of temporal logic are quite important in some domain like maintenance tasks or repeated tasks that require the agent conduct task periodically.
Therefore, the current paper overclaims what types of STL the proposed method can handle. The paper is supposed to have a clear explanation on what the specific STLs are they are targeting, give a formal definition on what these STLs are and what common properties they have in common, how they are related to other types of STLs that the method cannot handle.

The proposed reward function lacks a direct link with respect to the original STL task. It is unclear under this combination of rewards, how the optimal policy behaves in the original STL task. Ideally, the conversion from STL task to RL task should be formulated in a way that both optimization tasks share the same optimal solutions. Similar to LTL2Action, it would be better to provide analysis showing that the optimal policy under the proposed rewards also guarantees the satisfaction of the original STL.

**Questions:**

How are the STLs designed?

What are the guidelines used for this designing?

Are there any practical scenarios where these STLs have been applied in real applications?

---

> ### Author Response · Authors · 2025-11-21
>
> Thank you for recognizing the novelty of our work. There is limited literature addressing the STL planning problem with an emphasis on temporal aspects, which are crucial yet often overlooked by the community. Below, we provide detailed discussions for the weaknesses (W) and questions (Q) raised.
>
> > W1: Generability
>
> Our method focuses on finite-horizon STL with bounded time intervals, and we have clarified this scope in the revised version (Section 1, last paragraph). While formulas like $G(F(\cdots))$ might not be efficiently handled by TGPO, we respectfully clarify that TGPO has no problem in handling liveness properties. The Figure 1 in our paper illustrates how we solve multiple liveness properties, and Figure 2 shows how we solve nested liveness properties (other STLs with multiple liveness properties are shown in Appendix A.10.1-A.10.5). The ability to handle deep, nested STLs is one of the main advantages of TGPO. We have mentioned the STLs we can solve in the original manuscript (Section 4.1,  "In this work, we do not consider ...") To re-emphasize the class of the STL tasks we can solve, we have revised the paper to state the scope (Section 3.1, last paragraph).
>
>
> > W2: Theoretical analysis
>
> We added a theoretical analysis in the revised paper (appendix A.5). We show that with a proper scaling for the reward terms, the optimal policy will satisfy the STL task, if the original problem is feasible.
>
> > Q1: How are the STLs designed? Guidelines?
>
> The STL formulas used for each scenario are shown in the appendix A.10 of our paper and they are largely inspired by references [1,2]. We construct predicates from task-related spatial requirements and compose them into bounded temporal operators. To challenge our framework, we include nested, long-horizon, and structurally complex STL tasks that go beyond simple reach-avoid formulas.
>
>
> > Q2: Any practical scenarios for these STLs in real applications?
>
> For the STLs formulas in our paper to test TGPO, their common structure (conjunctions of "eventually" and "always" operators; timed goal reaching, obstacle avoidance, and ordered mission execution) is representative of many real-world tasks such as autonomous driving [3] and smart cities [4].
>
>
> **We hope our response has addressed your concerns. We kindly ask if there are any remaining issues preventing a more positive evaluation, and we would like to continue the discussion in the remaining rebuttal period.**
>
>
> **References:**
> 1. Sun, Dawei, et al. "Multi-agent motion planning from signal temporal logic specifications." IEEE Robotics and Automation Letters 7.2 (2022): 3451-3458.
> 2. Meng, Yue, and Chuchu Fan. "TeLoGraF: Temporal Logic Planning via Graph-encoded Flow Matching." arXiv preprint arXiv:2505.00562 (2025).
> 3. Zhong, Ziyuan, et al. "Guided conditional diffusion for controllable traffic simulation." arXiv preprint arXiv:2210.17366 (2022).
> 4. Ma, Meiyi, et al. "STLnet: Signal temporal logic enforced multivariate recurrent neural networks." Advances in Neural Information Processing Systems 33 (2020): 14604-14614.

---

### Author Response · Authors · 2025-11-21
**Official Summary by Authors**

We thank all reviewers for their constructive feedback. We are glad to see **the reviewers reached consensus that our TGPO is novel and our empirical evaluation is sufficient, broad, and solid.**

We have uploaded a revised manuscript with changes marked in $\textcolor{OrangeRed}{\text{this color}}$. Here is a list of key revisions:

- **Clarifications on scope:** We refined Section 1 and Section 3 to clearly state the scope of STL formulas we address (finite-horizon, nested conjunctions).

- **Algorithm details:** We included the pseudocode for the symbolic STL decomposition (Section 4.1, Algorithm 1).

- **Theoretical analysis:** We added a formal analysis in Appendix A.5 to establish the link between our dense reward design and STL satisfaction. We prove that under specific reward conditions, maximizing the expected return is equivalent to maximizing the STL success rate.

- **Additional experiment:** We conducted a varied-horizon test for the Unicycle dynamics (Appendix A.6) to demonstrate the consistency of our method across different dynamics.

We have addressed each reviewer's specific concerns in the separate threads below. **Happy to discuss further for any remaining questions during the rebuttal phase.**

---

> ### Author Response · Authors · 2025-11-26
>
> We want to gently follow up to ensure the reviewers have had a chance to review our rebuttal (regarding scope, additional experiments, and theoretical analysis). If there are any outstanding concerns that might hinder a more positive assessment, please let us know; we would greatly value the opportunity to resolve them. Thanks!

---

### Meta-Review · Area_Chair_yj7n · 2026-01-05

**Summary:**

This meta-review recommends a Reject for the submission. The primary concerns informing this decision center on the restricted scope of the Signal Temporal Logic (STL) fragment addressed and the scalability of the proposed temporal grounding mechanism. While the reviewers generally appreciated the novelty of using Metropolis-Hastings sampling to handle temporal assignments and the empirical performance across diverse environments, the consensus shifted toward a more critical view of the paper's theoretical and general applicability. Specifically, the method’s inability to handle fundamental STL operators like disjunctions and liveness properties (e.g., always eventually) suggests that the paper over-claims its ability to solve general STL tasks. Furthermore, the exponential growth of the temporal assignment search space relative to the number of subgoals remains a significant technical hurdle that limits the framework’s utility for truly complex, long-horizon specifications.

**Reviewer Concerns:**

During the rebuttal, the authors were proactive in addressing several technical gaps. They provided a theoretical analysis in Appendix A.5 to link the dense reward function to STL satisfaction—a point of disagreement for Reviewers G7gi and TtqM. They also clarified the problem statement regarding initial state distributions and improved the algorithmic description of the STL decomposition. However, several critical issues remain outstanding. Reviewer TtqM correctly identified that while the authors removed the word "general" from their claims, the current fragment of STL supported is quite narrow. The authors' defense of scalability—relying on the "density of feasible solutions" and empirical results—is more intuitive than rigorous. The reliance on a critic-guided heuristic for Bayesian sampling, while innovative, does not fully alleviate the fundamental combinatorial complexity inherent in the temporal decomposition for tasks with high subgoal density.

**Reviewer Scores:**

The divergence between the high score from Reviewer sqA6 and the low score from Reviewer TtqM is telling of the confidence levels. Reviewer sqA6, despite their positive rating, admitted to low confidence and likely overlooked the technical limitations of the STL fragment. Had they participated in the full technical discussion with TtqM, their score would likely have drifted toward a 5 or 6 as the limitations became clear. Reviewer G7gi remained skeptical of the paper's generality; while they might have moved to a 5 following the inclusion of the reward-satisfaction proof, the limited scope would have prevented an "Accept" recommendation. Reviewer TtqM maintained their 2, and given their detailed follow-up, it is unlikely they would have changed their stance without a more fundamental expansion of the algorithm's capabilities. Consequently, the lack of support from high-confidence reviewers leads to a rejection.

---

### Decision · Program_Chairs · 2026-01-26

Reject